# Rock Glacier Inventories (RoGI) in 12 areas worldwide using a multi-operator consensus-based procedure

Line Rouyet[1,2], Tobias Bolch[3,4], Francesco Brardinoni[5], Rafael Caduff[6], Diego Cusicanqui[7], Margaret Darrow[8], Reynald Delaloye[1], Thomas Echelard[1,5], Christophe Lambiel[9], Cécile Pellet[1], Lucas Ruiz[10], Lea Schmid[1], Flavius Sirbu[11], Tazio Strozzi[6]

[1]Dept. of Geosciences, University of Fribourg (UNIFR), Fribourg, 1700, Switzerland.
[2]NORCE Norwegian Research Centre AS, Tromsø, 9294, Norway.
[3]Institute of Geodesy, Graz University of Technology (TU Graz), Graz, 8010, Austria.
[4]Central-Asian Regional Glaciological Centre of Category 2 under the auspices of UNESCO, Almaty, Kazakhstan
[5]Dept. of Biological, Geological, and Environmental Sciences, University of Bologna (UniBo), Bologna, 40126, Italy.
[6]GAMMA Remote Sensing AG, Gümligen, 3076, Switzerland.
[7]Institut des Sciences de la Terre (ISTerre), Université Grenoble Alpes (UGA), Saint-Martin-d'Hères, 38400, France.
[8]Dept. of Civil, Geological, and Environmental Engineering, University of Alaska Fairbanks (UAF), Fairbanks, AK 99775-5900, United Stated of America.
[9]Institute of Earth Surface Dynamics, University of Lausanne (UNIL), Lausanne, 1015, Switzerland.
[10]Argentine Institute of Nivology, Glaciology and Environmental Sciences (IANIGLA), Mendoza, 5500, Argentina.
[11]Institute of Advanced Environmental Research, West University of Timișoara (WUT), Timișoara, 300223, Romania.

Correspondence to: Line Rouyet (line.rouyet@unifr.ch, liro@norceresearch.no)

**Abstract.** The Rock Glacier Inventories and Kinematics community (RGIK) has defined standards for generating Rock Glacier Inventories (RoGI). In the framework of the European Space Agency Climate Change Initiative for Permafrost (ESA CCI Permafrost), we set up a multi-operator mapping exercise in 12 areas around the world. Each RoGI team was composed of five to ten operators, involving 41 persons in total. Each operator performed similar steps following the RGIK guidelines (RGIK, 2023a) and using a similar QGIS tool. The individual results were compared and combined after common meetings to agree on the final consensus-based solutions. In total, 337 "certain" rock glaciers have been identified and characterised, and 222 additional landforms have been identified as "uncertain" rock glaciers.

The dataset consists of three GeoPackage files for each area: 1) the Primary Markers (PM) locating and characterising the identified Rock Glacier Units (RGU), 2) the Moving Areas (MA) delineating areas with surface movement associated with the rock glacier creep, based on spaceborne Interferometric Synthetic Aperture Radar (InSAR), and 3) the Geomorphological Outlines (GO) delineating the restricted and extended RGU boundaries. Here we present the procedure for generating consensus-based RoGI, describe the data properties, highlight their value and limitations, and discuss potential applications. The final PM/MA/GO dataset is available on Zenodo (Rouyet et al., 2025; https://doi.org/10.5281/zenodo.14501398). The GeoPackage (gpkg) templates for performing similar RoGI in other areas, and exercises based on the QGIS tool, are available on the RGIK website (https://www.rgik.org).

# 1    Introduction

Permafrost is defined as subsurface material remaining at or below 0 °C for at least two consecutive years (French, 2007). Due to its sensitivity to climate change, permafrost is an Essential Climate Variable (ECV), traditionally documented by the Ground Temperature (GT) and the Active Layer Thickness (ALT) (Streletskiy et al., 2017). In mountains, the permafrost distribution may be discontinuous and controlled by site-specific conditions with large variations over short distances. The investigation of mountain permafrost requires the development of dedicated products to complement to GT and ALT

measurements and models. Rock glaciers are obvious expressions of mountain permafrost, defined as debris landforms generated by the former or current creep of frozen ground (RGIK, 2023a). Although contrasting views exist in the genetic origin of rock glaciers, the distribution of rock glaciers may be regarded as a proxy of past or present permafrost occurrence. Rock glacier inventories (RoGI), including relict, transitional, and active landforms, are valuable to understand the evolution of periglacial environments, and to calibrate or validate mountain permafrost distribution models, where in situ

measurements are scarce (Azócar et al., 2017; Boeckli et al., 2012; Etzelmüller et al., 2020; Karjalainen et al., 2020; Marcer et al., 2017; Schmid et al., 2015). The distribution, sizes and dynamics of rock glaciers also have several operational implications for the management of geohazards and water resources, which have justified RoGI compilation in many mountain ranges (Hassan et al., 2021; Jones et al., 2018; Marcer et al., 2019; Rangecroft et al., 2015).

In addition, rock glacier creep rate is influenced by the permafrost thermal state and the ground ice/water contents (Cicoira et

al., 2019; Ikeda et al., 2008; Kenner et al., 2020). Several studies demonstrated that the interannual rock glacier velocity changes relate to the ground temperature variations (Delaloye et al., 2008; 2010; Kääb et al., 2007; Kellerer-Pirklbauer et al., 2024; Schoeneich et al., 2015; Staub et al., 2016). In the context of climate change, cases of acceleration, destabilisation, and even collapse have been reported (Bodin et al., 2017; Delaloye et al., 2013; Eriksen et al., 2018; Hartl et al., 2023; Kellerer-Pirklbauer et al., 2024; Scotti et al., 2017). Conversely, as degradation continues, rock glaciers tend to decelerate and

transition progressively into relict landforms (Ikeda & Metsuoka, 2002; Manchado et al., 2024; Necsoiu et al., 2016). Due to the link between temperature and rock glacier creep rate, Rock Glacier Velocity (RGV) became a new product of the ECV Permafrost (Hu et al., 2025; Streletskiy et al., 2021; WMO, 2022). In this context, RoGI compilation can be considered as a first necessary step to identify and select landforms to be monitored in a climate-oriented perspective. However, RoGI are not exhaustive worldwide and existing RoGI have been compiled with various methodologies. Owing to a lack of concerted

international rules for mapping and characterising rock glaciers, a RoGI compiled by different operators may lead to high levels of variability (Brardinoni et al., 2019), which hampers our ability to compare, merge, and analyse inventories across different regions.

With these motivations, the Rock Glacier Inventories and Kinematics (RGIK) initiative, launched in 2018, has focused on defining widely accepted standards and developing guidelines for the generation of RoGI and RGV products (Delaloye et al.,

2018). With the long-term objective to generate a homogenous open-access RoGI database, RGIK has released RoGI guidelines defining rules for inventory rock glaciers (RGIK, 2023a). In parallel, the European Space Agency Climate

Change Initiative for Permafrost (ESA CCI Permafrost) has worked on scaling up the generation and evaluation of ECV permafrost products using satellite remote sensing (Bartsch et al., 2023; Trofaier et al., 2017). For rock glacier products, ESA CCI Permafrost especially focuses on the use of spaceborne Interferometric Synthetic Aperture Radar (InSAR), an established remote sensing technique documenting ground surface movement and widely applied in the RoGI framework (Bertone et al., 2024; Brencher et al., 2021; Hu et al., 2023; Lambiel et al., 2023; Liu et al., 2013; Ma & Oguchi, 2024; Reinosch et al., 2021; Rouyet et al., 2021).

Previous studies highlighted the inherent subjectivity of operators to interpret the morpho-kinematic characteristics of rock glaciers based on optical and InSAR data (Bertone et al., 2022) and the benefits of designing multi-operator consensus-based procedures to reduce discrepancies and improve the final products (Way et al., 2021). In 2023, we therefore designed a mapping exercise with teams including operators from diverse institutions, countries, and backgrounds. This multi-operator RoGI exercise was performed in 12 areas around the world. Several operators performed similar steps individually and then discussed the results to provide consensus-based final products. This unique international initiative had four main objectives: 1) train the community for RoGI production, 2) test common RoGI rules and identify discrepancies to refine the existing guidelines, 3) develop standardised GIS templates and training tools for enhancing the production of comparable RoGI in new regions, and 4) compile and disseminate a homogenised set of RoGI from 12 diverse regions.

Here we present the multi-operator inventorying procedure (Section 2), describe the GIS tool and data properties (Section 3), summarise the main characteristics of the resulting dataset (Section 4), discuss the uncertainties and limitations (Section 5), and suggest ideas for future use and applications (Section 6).

## 2 Multi-operator inventorying procedure

### 2.1 RoGI areas and teams

The exercise was performed in 12 areas selected in ten countries and five continents (Table 1; Figure 1). Most RoGI areas have been selected within larger regions previously studied by Bertone et al. (2022), who included detailed descriptions of the regional settings in the supplementary material of the article. A Principal Investigator (PI) was designated to coordinate the work of the inventory team in each area. All PIs had past or ongoing research in the area they were leading. The volunteer operators were found within the involved institutions and after a call for participation in June 2023 using the RGIK mailing list (about 200 subscribers). The participants were free to choose one or more area(s) to perform the work, depending on their interest and time availability. To ensure enough operators in each area, as well as a diversity of geographical background, competence and seniority, members of the PI team acted as operators in areas where few people signed up. The resulting inventory teams were composed of five to ten operators (including the PI; Table 1). Some operators worked in several areas. One operator (R. Delaloye) performed the work in all the areas, which helped communicating common challenges and coordinating key decisions across the teams. The exercise involved a total of 41 persons (see Author list and Acknowledgments).

**Table 1. RoGI areas and teams (PI acronyms: see author list and affiliations).**

| Area number (ESA CCI Permafrost convention) | Area name (country, code) / Approx. central lat./long. location | AOI km$^2$ / Elevation range | PI (institution) / (# operators, incl. the PI) |
|---|---|---|---|
| Area 5-1 | Carpathians (Romania, RO) / 45°23' N, 22°53' E | 18 / ~1070 to ~2500 m a.s.l. | FS / WUT / (7 operators) |
| Area 6-1 | Western Alps (Switzerland, CH) / 46°11' N, 7°30' E | 12 / ~2160 to ~3000 m a.s.l. | TE / UNIFR / (5 operators) |
| Area 7-1 | Troms (Norway, NO-T) / 69°23' N, 20°26' E | 47 / ~400 to ~1400 m a.s.l. | LRo / NORCE / (6 operators) |
| Area 8-1 | Finnmark (Norway, NO-F) / 70°45' N, 27°50' E | 15 / 0 to ~535 m a.s.l. | LRo / NORCE / (7 operators) |
| Area 9-1 | Nordenskiöld Land (Norway, NO-N) / 77°53' N, 13°54' E | 10 / ~50 to ~900 m a.s.l. | LRo / NORCE / (6 operators) |
| Area 10-1 | Vanoise Massif (France, FR) / 45°19' N, 6°37' E | 37 / ~1710 to ~3150 m a.s.l. | DC / USMB/UGA / (6 operators) |
| Area 11-1 | Southern Venosta (Italy, IT) / 46°33' N, 10°36' E | 19 / ~2120 to ~3545 m a.s.l. | FB / UniBo / (10 operators) |
| Area 12-1 | Disko Island (Greenland, GL) / 69°51' N, 52°33' W | 82 / 0 to ~1330 m a.s.l. | RC / GAMMA / (6 operators) |
| Area 13-1 | Northern Tien Shan (Kazakhstan, KA) / 43°0' N, 77°1' W | 59 / ~2570 to ~4365 m a.s.l. | TB / TU Graz / (7 operators) |
| Area 14-1 | Brooks Range (Alaska, U.S.A., US) / 68°6' N, 149°58' W | 21 / ~1120 to ~2070 m a.s.l. | MD / UAF / (10 operators) |
| Area 15-1 | Central Andes (Argentina, AR) / 32°59' S, 69.34° W | 55 / ~3570 to ~5530 m a.s.l. | LRu / IANIGLA / (10 operators) |
| Area 16-1 | Southern Alps (New Zealand, NZ) / 43°59' S, 170°3' E | 7 / ~1600 to ~2431 m a.s.l. | CL / UNIL / (7 operators) |

Rouyet et al. ESSD revised manuscript 19/05/2025

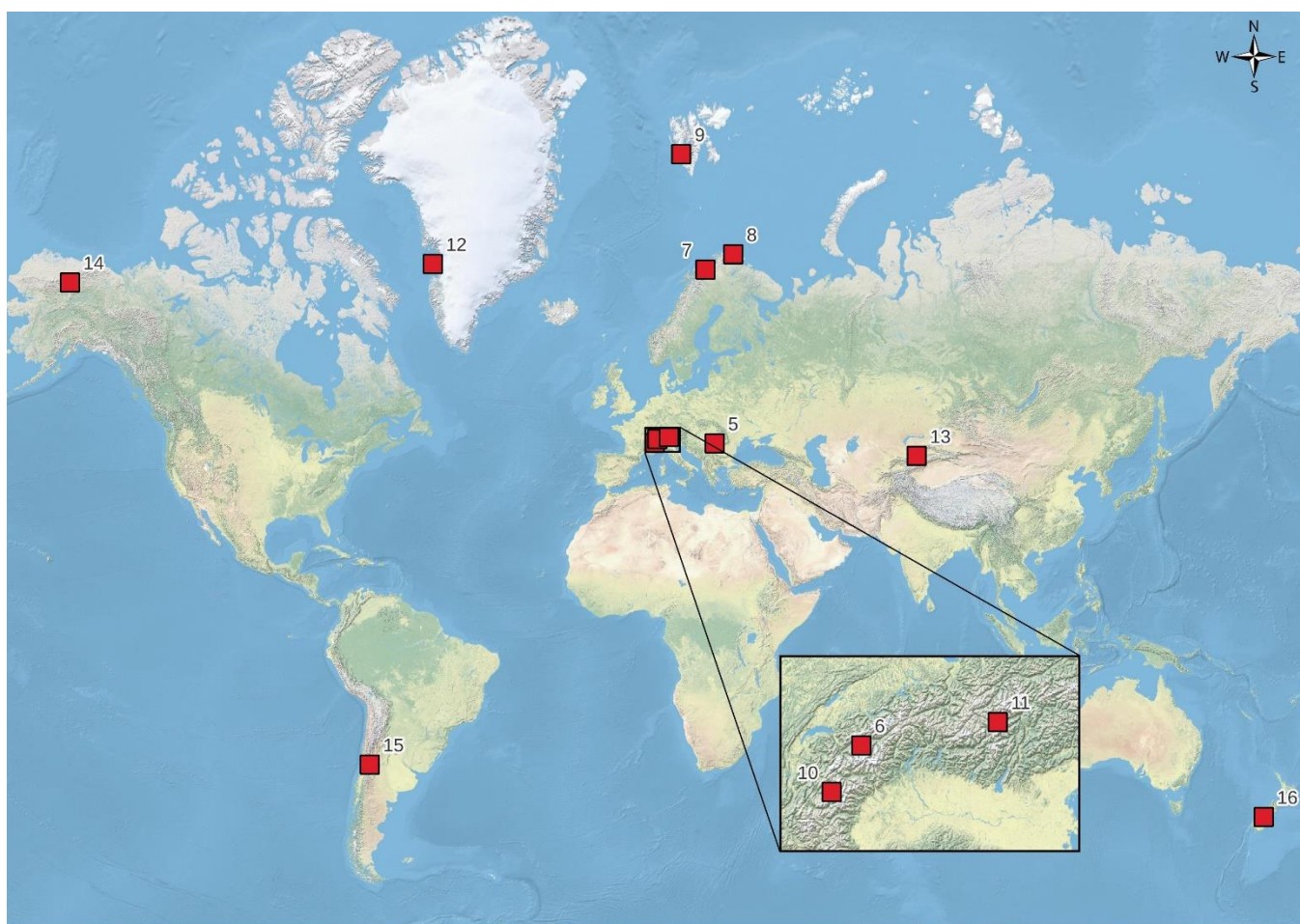

**Figure 1. Location map of the RoGI regions, including the areas selected for the multi-operator RoGI exercise. Background map: ESRI Physical Web Map Service.**

## 2.2 Consensus-based RoGI procedure

The RoGI exercise was performed between June and November 2023. The University of Fribourg (UNIFR), Switzerland, was responsible for providing the data packages and instructions, and coordinating the work between the 12 teams, corresponding to the 12 areas. For each area, the PI coordinated the work and had the responsibility for the final products. The PI also performed the work as an operator. Within each team, each operator received a common folder including a similar dataset applicable for the area. The data is organised within a QGIS project (see Section 3.1), along with the instructions for the exercise and the references to the RGIK guidelines applicable at the time (RGIK, 2022a; 2022b; 2022c; 2023b). The guidelines have since been merged into one reference document (RGIK, 2023a).

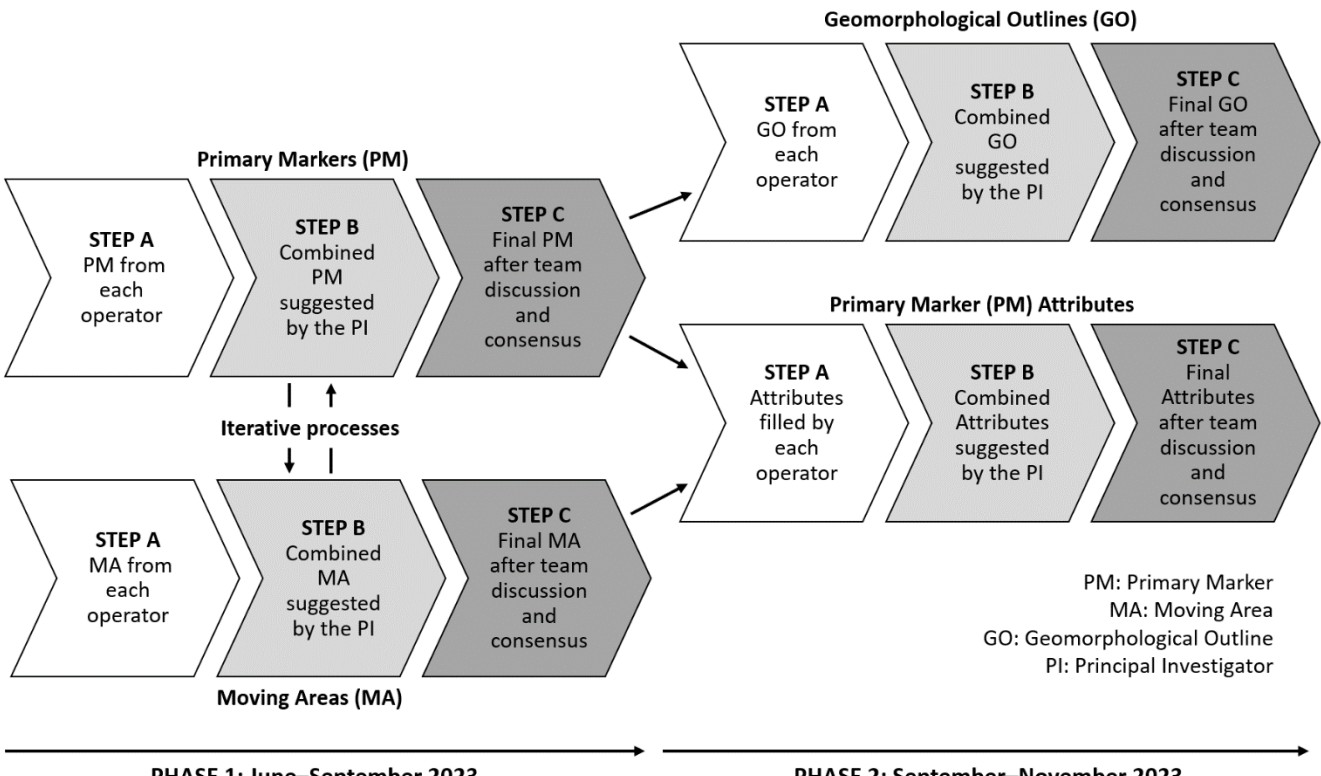

**Figure 2. Consensus-based RoGI procedure.**

The inventory procedure included two main phases, performed in June–September 2023 (Phase 1) and in September–November 2023 (Phase 2) (Figure 2). Each phase was divided into three steps:

- **Step A**: Individual work by each team operator. At the end of this step, all the operators sent their results to the PI.
- **Step B**: Compilation and summary by the PI. When discrepancies between operators were identified, the PI suggested a solution, to be discussed with the team.
- **Step C**: Discussion and consensus-based final decision by the inventory team. At the end of this step, the team agreed to the intermediate (first phase) or the final outputs (second phase).

During the first phase performed between June and September 2023, the team had to:

- **Identify and locate the Rock Glacier Units (RGU) with Primary Markers (PM)**. The operators were asked to include landforms following the technical definition of a rock glacier: "a debris landforms generated by the former or current creep of frozen ground, detectable in the landscape with the following morphologies: front, lateral margins and optionally ridge-and-furrow surface topography" (RGIK, 2023a, p.6). Based on this definition, a RoGI must include relict rock glaciers, but discard landforms that are primarily driven by other processes, such as glacial flow, solifluction,

ice melt, and sliding along a slip-surface. Different units are discriminated according to the RGIK guidelines (RGIK, 2023a). Orthoimages were the primary source of data used for this task, but additional datasets were used when available (e.g., Digital Elevation Model, DEM) (see Section 3.1). InSAR data was useful to detect or confirm the location of moving rock glaciers. Each recognised RGU was identified with a point (primary marker, PM) in a dedicated vector layer. An uncertainty could be expressed by defining the landform as "uncertain rock glacier" in the case of geomorphological ambiguity or low data quality. When combining the results between operators, the team agreed on which units were categorised as "certain" or "uncertain" within each area. In some cases, the rock glaciers remained "uncertain" when there was not enough evidence that the landform is a rock glacier, or the teams decided that the landform was too complex to be accurately characterised and outlined with the currently available data. Keeping an information about the location of these uncertain landforms may allow for future updates if new data is becoming available. The operators could optionally use a label "not a rock glacier" to indicate landforms that may be mistaken for rock glaciers, but are not driven by permafrost creep. These complex cases were discussed during team meetings and sometimes kept in the final layer for educational purposes. The attribute table of the PM layer is shown in Appendix A. At this stage, only the first attributes of the table were applicable, as the detailed morpho-kinematic characterisation was performed during the second stage.

- **Detect, delineate, and classify Moving Areas (MA) using InSAR**. This task was performed in parallel, potentially iteratively, with the first bullet point (RGU identification with PM). The MA were identified, delineated, and characterised based on InSAR data (see Section 3.1). For each area, the operators used a similar collection of radar image pairs (interferograms) from different spaceborne radar sensors, with different viewing geometries and variable time intervals between the image acquisitions. In some areas, multi-temporal InSAR mean velocity maps based on Distributed Scatterer (DS) and Persistent Scatterer (PS) algorithms were also available (Table 2). Each recognised MA was delineated in a dedicated polygon vector layer. The attributes documenting the velocity class, the observation time window and validity time frame, and the MA reliability could be filled using a semi-automatic dialog box. The attribute table of the MA layer is shown in Appendix B. The boundaries of the MA polygons follow the InSAR signal, not the landform features. If the movement is heterogenous and/or if InSAR is affected by limitations, the MA may only be partly overlapping with the rock glacier. The MA step was performed before the team decisions on the RGU final locations, which means that some delineated MA may correspond to surface movement associated to uncertain rock glaciers or other periglacial processes. Such polygons were kept in the final layer but were not further used for morpho-kinematic characterisation when they did not correspond to a certain RGU. If no movement was detected on InSAR, no polygon was drawn. Several rock glaciers have therefore no corresponding MA. The complete procedure is explained in the RGIK practical InSAR guidelines (RGIK, 2023b).

In September 2023, the PI compared the individual results and suggested final solutions. After discussion and adjustment during an online meeting with the team operators, the final consensus-based PM and MA layers were adopted.

During the second phase performed between September and November 2023, the team focused on the landforms categorised as "certain rock glaciers" in the final PM layer. For those landforms, the operators had to:

- **Document the RGU morpho-kinematic characteristics (Attributes).** The morpho-kinematic attributes characterising the RGUs were filled using a semi-automatic dialog box in the final consensus-based PM layer from the first phase. All attributes refer to definitions described in the RGIK RoGI guidelines (RGIK, 2023a). All documented attributes are listed in Appendix A. For the geomorphologic attributes, orthoimages were the primary source of data, but additional datasets were used (e.g., DEM) (see Section 3.1, Table 2). The kinematic attribute (KA) is a semi-quantitative estimate
of the overall multi-annual movement rate of the rock glacier unit (order of magnitude: cm/a, dm/a, m/a, etc), summarizing the information provided by the MA layer, when it overlaps with the identified rock glacier landforms. The procedure to convert velocity information from one or several MA polygons to one KA category is explained in the RGIK guidelines (RGIK, 2023a; 2023b). The KA was used to assess the activity (active, transitional, relict), defined as the efficiency of sediment conveyance (expressed by the surface movement).

- **Delineate the RGU Geomorphological Outlines (GO).** The extended and restricted rock glacier GO were delineated in a dedicated polygon vector layer. The extended outline embeds the entire rock glacier up to the rooting zone and include the external parts (front and lateral margins). The restricted outline embeds the entire rock glacier up to the rooting zone exclude the external parts (front and lateral margins) (RGIK, 2023a). For each polygon, attributes (outline type and reliability of the delineation) could be filled using a semi-automatic dialog box. The attribute table of the GO
layer is shown in Appendix C.

In November 2023, the PI compared the individual results and suggested final solutions. After discussion and adjustment during an online meeting with the team operators, the final consensus-based PM Attributes and the GO layer were adopted.

The compilation, data harmonization, and technical correction of the final set of PM, MA, and GO products were performed
by the University of Fribourg, Switzerland (UNIFR) between November 2023 and February 2024. A final verification and approval by the PIs was performed between February and May 2024. Thanks to the referee's comments on the first submission of the dataset and the associated paper, new corrections have been performed in March 2025.

## 3 Data types, attributes and formats

### 3.1 Input data and GIS tool

The data packages delivered to the operators all had the same structure. The content was similar for each area. The main folder included four subfolders and a QGIS project:

- **Subfolder "INSTRUCTIONS"** with the documents and links to the applicable guidelines.

- **Subfolder "VECTOR"** including the polygon of the Area of Interest (AOI) that defined the boundaries in which the inventory work had to be performed, as well as the initial geopackage (gpkg) templates for digitalising the PM, MA, and GO.

- **Subfolder "INSAR-DATA"** including wrapped interferograms from Sentinel-1 (and potentially ALOS, SAOCOM, Cosmo-SkyMed, and/or TerraSAR-X depending on the data availability), potential complementary InSAR products (e.g., velocity maps from 6–12 days Stacking, combined 6d–annual Stacking, and/or Persistent Scatterer Interferometry algorithms), a layer displaying an index to reproject the line-of-sight displacement rate along the direction of the steepest slope (normalization factor) or a mask highlighting N–S facing slopes where the InSAR data is likely to underestimate the real movement. These datasets are summarised in Table 2, and further explained in the InSAR guidelines (RGIK, 2023b).

- **Subfolder "DEM-ORTHO"** in which the PI could add extra available background data before delivery to the operators (e.g., DEM-based products, high-resolution orthophotos, topographic maps, see Table 2).

- **QGIS project** structuring the available data and in which the operators performed the work. In addition to the AOI, the InSAR data and initial vector files (gpkg templates), each GIS project incorporated links to Web Map Services (WMS) such as the Google Earth, Bing and ESRI orthomosaics (Table 2). The spatial resolution of such images is typically 0.1–1 m but varies within/across the areas and depending on the scale and zoom levels.

The work was performed in similar QGIS projects, with common file structure, background data, and dialog boxes for filling the attribute tables. The QGIS structure is generic and allows for semi-automatic attribute selection to simplify the work of the operators (Figure 3).

**Table 2: Summary of input data in each RoGI area. The names and locations corresponding to the area numbers are shown in Table 1. The crosses (x) highlight the availability of the corresponding dataset. For InSAR data: the yy-yy numbers correspond to the years available for each InSAR dataset (e.g., 15-19: interferograms or averaged velocity maps between 2015 and 2019).**

| Area number (see Table 1) | 5-1 | 6-1 | 7-1 | 8-1 | 9-1 | 10-1 | 11-1 | 12-1 | 13-1 | 14-1 | 15-1 | 16-1 |
|---|---|---|---|---|---|---|---|---|---|---|---|---|
| **Satellite Web Map Services (WMS): Optical imagery and topographical map** | | | | | | | | | | | | |
| Google satellite WMS | x | x | x | x | x | x | x | x | x | x | x | x |
| Bing satellite WMS | x | x | x | x | x | x | x | x | x | x | x | x |
| ESRI satellite WMS | x | x | x | x | x | x | x | x | x | x | x | x |
| OpenTopoMap WMS | x | x | x | x | x | x | x | x | x | x | x | x |
| **Additional optical/thematic data:** HR aerial imagery and national topographical map | | | | | | | | | | | | |
| Extra HR aerial image | x | x | x | x | x | x | | | | | | |
| National topo. map | | x | x | x | | x | | | | | | x |
| **DEM products:** Low/High-Resolution (LR/HR) DEM and/or associated products (e.g., hillshades, slope, aspect) | | | | | | | | | | | | |
| LR DEM (10–30m) | x | x | x | x | x | x | x | x | x | x | x | x |

Rouyet et al. ESSD revised manuscript 19/05/2025

| | | | | | | | | | | | | |
|---|---|---|---|---|---|---|---|---|---|---|---|---|
| HR DEM (< 10m) | x | x | | | | x | x | | | | | |
| **InSAR data:** Wrapped interferograms (ifgs) and velocity maps from Stacking and Persistent Scatterer Interferometry (PSI) | | | | | | | | | | | | |
| Sentinel-1 ifgs | 16-19 | 17-19 | 17-19 | 17-20 | 18-20 | 16-19 | 18-19 | 15-19 | 15-19 | 16-19 | 18-20 | 15-23 |
| ERS-1/2 ifgs | | | | | | | | | 98-99 | 91-95 | | |
| ALOS-1 ifgs | | | | | | 07-10 | 07-10 | | 06-10 | 06-09 | 08-11 | 07-08 |
| ALOS-2 ifgs | 14-19 | 14-21 | | | | | | 15-17 | 14-16 | 15-16 | 16-19 | |
| SAOCOM ifgs | | 21 | | | | | | | | | 21-22 | 21-23 |
| Cosmo-SkyMed ifgs | | | | | | | 16-20 | | | | | |
| TerraSAR-X ifgs | | 09-14 | | | | | | | | | | |
| 6–12d ifgs Stacking | | 19 | 15-19 | 15-20 | 15-20 | 18-19 | 18-19 | 18 | 18 | 18-19 | 18-19 | 18 |
| Combined 6d–annual ifgs Stacking | | | 15-19 | 15-20 | 15-20 | | | | | | | |
| PSI | 15-21 | | | 15-19 | | | | | | | | |

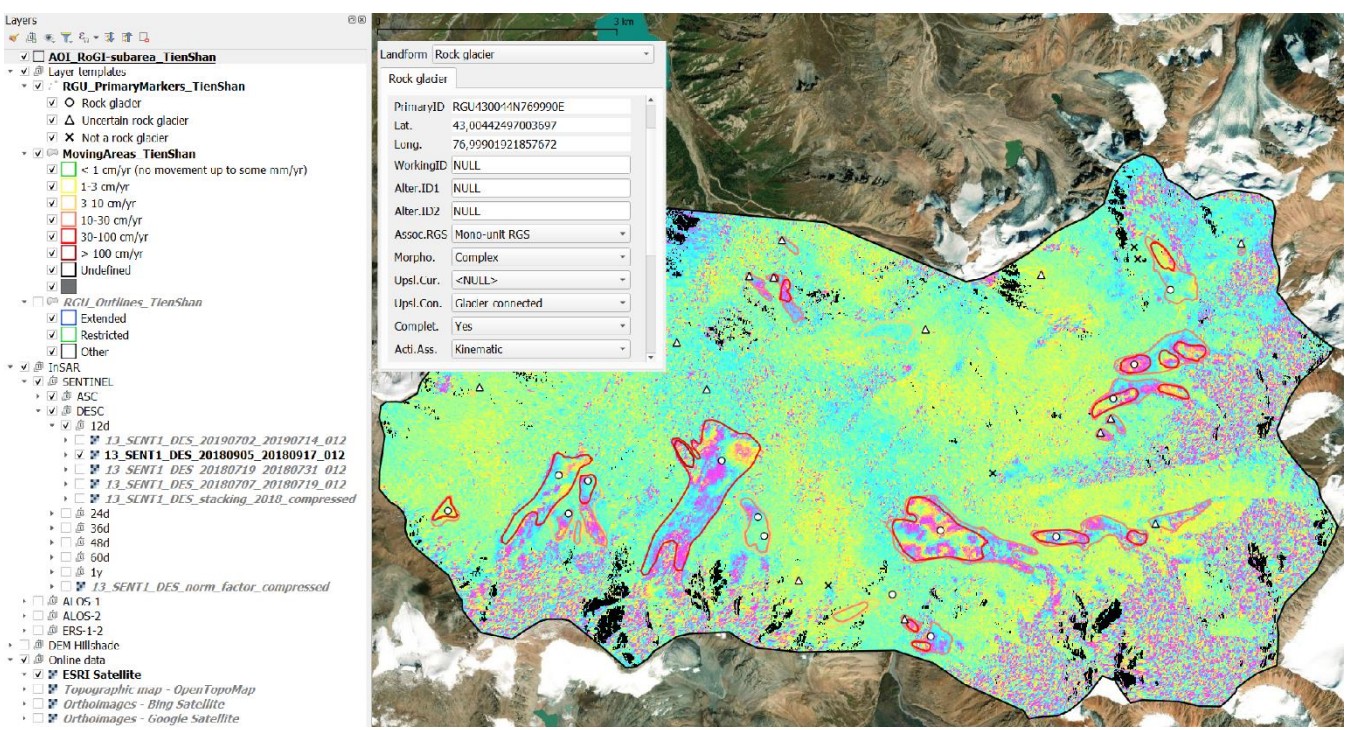

**Figure 3. Example of QGIS data structure and dialog box for semi-automatic attribute filling in area 13-1 (Northern Tien Shan, Kazakhstan). An example of Sentinel-1 wrapped interferogram is displayed within the AOI extent. The boundaries of the RoGI area (black polygon), the PM (white dots and triangles), and the MA (yellow to red polygons) are displayed as top layers. For sake of visualisation, the GO layer is not shown. See example with GO in Figure 4. Background map: ESRI Satellite Web Map Service.**

### 3.2 Output data: format and properties

The RoGI multi-operator exercise led to the generation of a set of three files for each area: the RGU Primary Markers (PM), the InSAR-based Moving Areas (MA), and the RGU Geomorphological Outlines (GO). All datasets are provided in a GeoPackage vector format (gpkg), a platform-independent database container. The Coordinate Reference System (CRS) used for the RoGI products is the World Geodetic System 1984 (WGS84). The coordinates are specified in decimal degrees.

**For the RGU Primary Markers (PM)**, the following attributes are documented:

- ID (unique alpha-numerical identifier of the RGU).
- X and Y coordinates (WGS84 coordinate system).
- Morphological type (simple, complex).
  Additional related attribute: the "Completeness" field defining if the rock glacier is completely visible or not (complete,
unclear connection to the upslope, truncated front, uncertain).
- Spatial connection to the upslope unit (talus-, debris mantle-, landslide-, glacier-, glacier forefield-, poly-connected, other, uncertain, unknown).
  Additional related attributes: the "Upslope current" field defining if the rock glacier is currently connected to the upslope unit or not, and a "Comment" field to further describe morphological characteristics.
- Kinematic attribute (< cm/yr, cm/yr, cm/yr to dm/yr, dm/yr, dm/yr to m/yr, m/yr, > m/yr, undefined).
  Additional related attributes: the "Type of Data" field to define the type of data used to assign the kinematic attribute (Optical, Radar, Lidar, Geodetic, Other), the "Kinematic Period" field to document the applicable period of the kinematic attribute (year(s) with available data), the "Reliability" of the kinematic attribute (low, medium, high, undefined), and a "Comment" field to document the applied method and the data quality.
- Activity (active, active uncertain, transitional, transitional uncertain, relict, relict uncertain, uncertain).
  Additional related attribute: the "Activity assessment" field documenting how the activity has been assessed (morphological evidence only or with kinematic data).
- Destabilisation signs (yes – ongoing, yes – completed, no, undefined).

**For the Moving Areas (MA)**, the following attributes are documented:

- ID (unique alpha-numerical identifier of the moving area)
- Velocity class (< 1 cm/yr, 1–3 cm/yr, 3–10 cm/yr, 10–30 cm/yr, 30–100 m/yr, >100 cm/yr).
- Time observation window (text documenting the time period used for the MA detection and characterisation).
- Reliability of the detected moving area (low, medium, high).
- Additional comments.

Rouyet et al. ESSD revised manuscript 19/05/2025

For the Geomorphological Outlines (GO), the following attributes are documented:

- ID (unique alpha-numerical identifier of the moving area)
- Outline type (extended, restricted, other).
- Reliability of the front, left margin, right margin, and upslope limit (0 – low, 1 – medium, 2 – high), and Reliability Index (automatic summation of the values assigned to the reliability attributes of these four different boundaries).
- Additional comments.

Each attribute is explained in detail in Appendixes A–C (including references to the applicable sections of the RGIK guidelines).

### 3.3  Output data: structure and naming convention

The data package is available on Zenodo (Rouyet et al., 2025; https://doi.org/10.5281/zenodo.14501398). It includes a set of gpkg files organised by areas and product types (PM, MA, GO).

The naming convention of each gpkg file follows the product specifications defined by the ESA CCI Permafrost project and is meant to provide a generic structure allowing for updates and/or release of future additional products. All file names follow the same structure: **ESACCI-<CCI Project>-<Processing Level>_<Data Type>_<Product String>-<Additional Segregator>_<Layer Type>_<Indicative Date>-fv<File version>.gpkg**

- **<CCI Project>:** PERMAFROST.
- **<Processing Level>**: Indicator (IND).
- **<Data Type>**: <SENSOR>-<METHOD>. <SENSOR> is the primary remote sensing data source used to document the kinematics, in this case: SENTINEL-1. <METHOD> is the primary method used to process the kinematic data, in this case: INSAR.
- **<Product String>**: ROGI, for the product Rock Glacier Inventory.
- **<Additional Segregator>:** This should be structured as: AREA_<REGION_NUMBER>-<AREA_NUMBER>. <REGION_NUMBER> follows the generic CCI Permafrost numbering: 5–Carpathians (Romania); 6–Western Alps (Switzerland); 7–Troms (Norway); 8–Finnmark (Norway); 9–Nordenskiöld Land (Svalbard, Norway); 10–Vanoise Massif (France); 11–Southern Venosta (Italy); 12–Disko Island (Greenland); 13–Northern Tien Shan (Kazakhstan); 14–Brooks Range (Alaska, U.S.A.); 15–Central Andes (Argentina), 16–Southern Alps (New Zealand). <AREA_NUMBER> is a one- or more-digit(s) number, depending on the numbers of area(s) in the region. For merged products (RoGI in all areas), the additional segregator is: ALL-AREAS.
- **<Layer Type>**: The individual layers of the vector product are provided in individual or merged files. The code of each individual layer is as follows:

- ▪ AOI: extent of the ROGI area.
- ▪ PM: layer 1, corresponding to the Primary Markers of the Rock Glacier Units.
- ▪ MA: layer 2, corresponding to the InSAR-based Moving Areas.
- ▪ GO: layer 3, corresponding to the Geomorphological Outlines of the Rock Glacier Units.

  The merged data package combining the different layers includes the three codes (PM-MA-GO).

- • **<Indicative Date>**: Format is YYYYMMDD, where YYYY is the year, MM is the month from 01 to 12, and DD is the
295 day of the month from 01 to 31. Annual or multi-annual products are represented with YYYY only.

- • **fv<File Version>**: File version number in the form n{1,}[.n{1,}] (two digits followed by a point and one or more digits).

Accordingly, the data package is structured as followed:

- • **The folder 'ESACCI-PERMAFROST_ROGI_SINGLE-AREA'**, including the RoGI products for each area, for applications focusing on one specific region, with subfolders named as follows:

  - • AREA_<AREA_NUMBER>_<AREA_NAME>_<COUNTRY_CODE>

    *Example: AREA_05-1_Carpathians_RO*

    - ▪ AOI, in a polygon vector layer in a gpkg format.
*Example: ESACCI-PERMAFROST-IND_SENTINEL1-INSAR_ROGI-AREA_5-1_AOI_2025-fv02.0.gpkg*

    - ▪ Primary Markers (PM), in a point vector layer in a gpkg format.

      *Example: ESACCI-PERMAFROST-IND_SENTINEL1-INSAR_ROGI-AREA_5-1_PM_2025-fv02.0.gpkg*

    - ▪ Moving Areas (MA), in a polygon vector layer in a gpkg format.

      *Example: ESACCI-PERMAFROST-IND_SENTINEL1-INSAR_ROGI-AREA_5-1_MA_2025-fv02.0.gpkg*

- ▪ Geomorphological Outlines (GO), in a polygon vector layer in a gpkg format.

      *Example: ESACCI-PERMAFROST-IND_SENTINEL1-INSAR_ROGI-AREA_5-1_GO_2025-fv02.0.gpkg*

- • **The file 'ESACCI-PERMAFROST_ROGI_ALL-AREAS_AOI-PM-MA-GO_2025_fv02.0.gpkg'**, including the AOIs and RoGI results (PM, MA and GO), merged for all areas, for applications requiring the combined use of all inventories.

- • **The file 'AAA_README_FIRST.pdf' file**, describing the data structure and properties.

## 4 RoGI results description

Figure 4 is an example of results of the RoGI multi-operator exercise for a selected area. It illustrates the similarities and differences between individual operator results (black dots for RGU PM; dashed lines for RGU GO) and the final products (coloured dots for RGU PM; solid lines for RGU GO). Due to the iterative and consensus-based procedure described in Section 2, the outcome is more than the sum of the individual results. The data package therefore includes the final consensus-based products only. In the following, we describe the results in each area separately (Sections 4.1–4.12) before summarising the findings across all areas (Section 4.13).

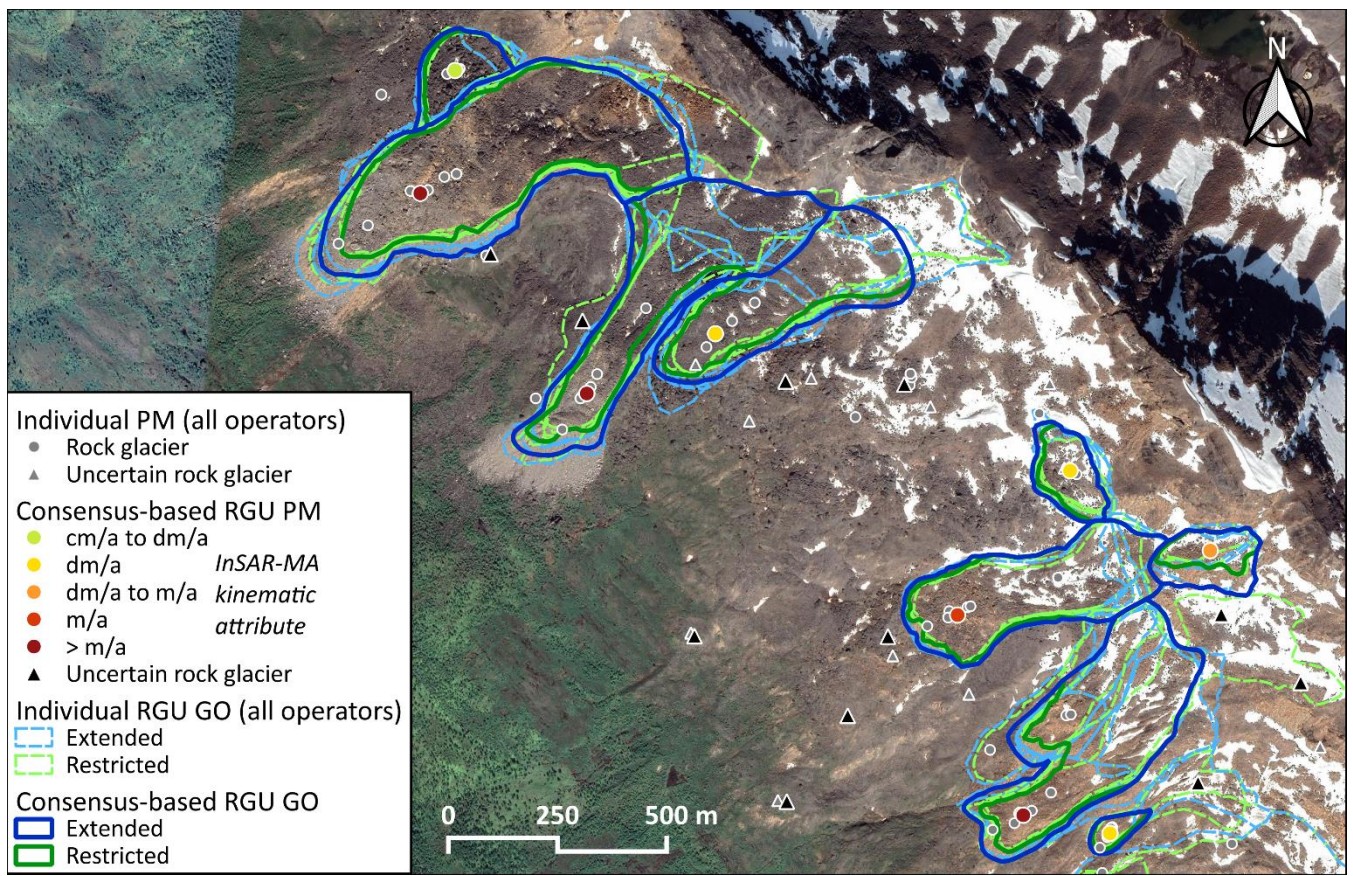

**Figure 4: Example of RoGI results in part of area 7-1 NO-T (Troms, Norway), showing individual operator results and final consensus-based results (Primary Markers: PM; Geomorphological Outlines: GO). For sake of visualisation, the MA layer is not shown, but was used to assign the PM kinematic attribute displayed here with a green−red colour scale. See example with MA in Figure 3. Background: NorgeiBilde orthophoto (2016-08-2016).**

### 4.1 RoGI area 5-1 RO (Carpathians, Romania)

RoGI area 5-1 is located in the Southern Carpathians, in Romania (central lat./long. location: 45°23' N, 22°53' E). The area covers an extent of approx. 18 km$^2$. The elevation ranges from peaks up to ~2500 m a.s.l along the southern mountain ridge, down to ~1070 m a.s.l. in the valley further north.

Previous research showed sporadic and isolated patches of permafrost, that are strongly linked with rock glaciers (Ardelean et al., 2015; Onaca et al., 2015; Popescu et al., 2024) and classified a small number of rock glaciers as active, with displacement rates on the order of cm/yr for the past decades (Necsoiu et al., 2016).

The multi-operator RoGI exercise resulted in the identification of 18 certain rock glacier units, and 11 uncertain features. The InSAR-based MA indicate velocities ranging from < 1 cm/yr to 3–10 cm/yr. The assigned KA has contributed to 340 classify the RGU activity as relict (12 RGU), and transitional (6 RGU). The averaged size of the mapped rock glaciers is ~0.07 km$^2$ based on the extended outlines.

### 4.2 RoGI area 6-1 CH (Western Alps, Switzerland)

RoGI area 6-1 is located in the upper part of the Réchy valley, in the Western Swiss Alps (central lat./long. location: 46°11' N, 7°30' E). The area covers an extent of approx. 12 km$^2$. The elevation ranges from peaks up to ~3000 m a.s.l along the 345 southern mountain ridge, down to ~2160 m a.s.l. in the valley further north.

Permafrost is still present in the upper part of the study area, whilst the lower area is mainly dominated by relict rock glaciers (Lugon & Delaloye 2001; Marthaler et al. 2008; Tenthorey 1992). The kinematics of the Becs-de-Bosson rock glacier has intensively been monitored since the early 2000s (Kellerer-Pirklbauer et al., 2024; PERMOS 2024; Perruchoud & Delaloye, 2007) and displays velocities up to 2 m/yr. Staub et al. (2016) used this site to evidence the dependency of the interannual 350 variation of the rock glacier creep rate to the multi-year ground surface temperature forcing.

The multi-operator RoGI exercise resulted in the identification of 30 certain rock glacier units, and 18 uncertain features. The InSAR-based MA indicate velocities ranging from 1–3 cm/yr to > 100 cm/yr. The assigned KA has contributed to classify the RGU activity as relict (23 RGU), transitional (4 RGU) and active (3 RGU). The averaged size of the mapped rock glaciers is ~0.03 km$^2$ based on the extended outlines.

### 4.3 RoGI area 7-1 NO-T (Troms, Norway)

RoGI area 7-1 is located in the Kåfjord–Storfjord mountainous region, in Troms County, Northern Norway (central lat./long. location: 69°23' N, 20°26' E). The area covers an extent of approx. 47 km$^2$. The elevation ranges from peaks up to ~1400 m a.s.l along the main Ádjit mountain ridge, down to ~400 m a.s.l. along the Skibotn valley flanks.

Previous research in this area indicated that the combination of seasonal frost and sporadic–discontinuous permafrost 360 conditions in the region leads to a wide diversity of periglacial slope processes (Rouyet et al., 2021), including very high

velocity rock glaciers (Eriksen et al., 2018). The distribution of relict and active rock glaciers fits the extents of the modelled Holocene and present-day permafrost extent in the region (Lilleøren et al., 2012).

The multi-operator RoGI exercise resulted in the identification of 15 certain RGU, and 26 uncertain features. The InSAR-based MA indicate velocities ranging from < 1 cm/yr to > 100 cm/yr. The assigned KA has contributed to classify the RGU activity as transitional (4 RGU), active (10 RGU), and active uncertain (1 RGU). The averaged size of the mapped rock glaciers is ~0.03 km$^2$ based on the extended outlines.

### 4.4 RoGI area 8-1 NO-F (Finnmark, Norway)

RoGI area 8-1 is located along Store Skogfjorden and Hopfjorden, in Finnmark country, Northern Norway (central lat./long. location: 70°45' N, 27°50' E). The area covers an extent of approx. 15 km$^2$. The elevation ranges from peaks up to ~535 m a.s.l in the southeastern part, down the sea level along the fjord.

The area is located at the limit of the modelled regional permafrost extent (Gisnås et al., 2017). Although most past research has interpreted Finnmark rock glaciers as relict landforms (Lilleøren & Etzelmüller, 2011), a recent multi-methodological study suggests that some rock glaciers at sea-level are at a transitional stage (Lilleøren et al., 2022).

The multi-operator RoGI exercise resulted in the identification of 17 certain rock glacier units, and 21 uncertain features. The InSAR-based MA indicate velocities ranging from < 1 cm/yr to 30–100 cm/yr. The assigned KA has contributed to classify the RGU activity as relict (15 RGU) and relict uncertain (2 RGU). The averaged size of the mapped rock glaciers is ~0.05 km$^2$ based on the extended outlines.

### 4.5 RoGI area 9-1 NO-N (Nordenskiöld Land, Norway)

RoGI area 9-1 is located in the Western part of Nordenskiöld Land on Spitsbergen, the main island of Svalbard (central lat./long. location: 77°53' N, 13°54' E). The area covers an extent of approx. 10 km$^2$. The elevation ranges from peaks up to ~900 m a.s.l along the southeastern part of mountain ridge, down to ~50 m a.s.l. on the Nordenskiöldkysten strandflat.

Past rock glacier research in Svalbard identified low creep rates despite continuous permafrost and ice-rich conditions (Isaksen et al., 2000; Berthling et al., 1998). Along Nordenskiöldkysten, the apparent standstill of rock glaciers has been attributed to the low slope gradients where the rock glaciers flow onto the strandflat (Farbrot et al., 2005).

The multi-operator RoGI exercise resulted in the identification of 18 certain rock glacier units, and 9 uncertain features. The InSAR-based MA indicate velocities ranging from < 1 cm/yr to 30–100 cm/yr. The assigned KA has contributed to classify the RGU activity as relict uncertain (3 RGU), transitional (9 RGU), and active (6 RGU). The averaged size of the mapped rock glaciers is ~0.04 km$^2$ based on the extended outlines.

### 4.6 RoGI area 10-1 FR (Vanoise Massif, France)

RoGI area 10-1 is located in the Vanoise massif in France, in the Western European Alps (central lat./long. location: 45°19' N, 6°37' E). The area covers an extent of approx. 37 km$^2$. The elevation ranges from peaks up to ~3150 m a.s.l in the southern part, down to ~1710 m a.s.l in the valley further north.

Previous research in this area indicated that sporadic and discontinuous permafrost conditions in the region leads to a wide diversity and complexity of periglacial slope processes and several examples of rock glaciers destabilisation (Marcer et al., 2021).

The multi-operator RoGI exercise resulted in the identification of 49 certain rock glacier units, and 51 uncertain features. The InSAR-based MA indicate velocities ranging from 1–3 cm/yr to > 100 cm/yr. The assigned KA has contributed to classify the RGU activity as relict (8 RGU), relict uncertain (2 RGU), transitional (13 RGU), active (20 RGU), and active uncertain (6 RGU). The averaged size of the mapped rock glaciers is ~0.03 km$^2$ based on the extended outlines.

### 4.7 RoGI area 11-1 IT (Southern Venosta, Italy)

RoGI area 11-1 is located in Solda Valley (Suldental), a tributary valley of the Venosta Valley (Vinschgau), in western South Tyrol, Italy (central lat./long. location: 46°33' N, 10°36' E). The area hosts two hanging valleys and covers an extent of approx. 19 km$^2$. The elevation ranges from peaks up to ~3545 m a.s.l for Cima Vertana in the eastern divide, down to ~2120 m a.s.l further southwest.

According to a recently compiled geomorphological inventory, the area is characterised by the highest rock glacier density within South Tyrol (~ 1.1 #/km$^2$ against a regional average of 0.54 #/km$^2$) (Scotti et al., 2024). Subsequent integration of this geomorphological inventory with InSAR-based kinematic information across the Southern Venosta subregion led to detect 375 intact and 428 relict rock glaciers (Bertone et al., 2024). On average, the velocity of intact rock glaciers was found to increase linearly with elevation up to the 2600–2800 m band (where MAAT declines from about -1 to -2 °C), beyond which a kinematic plateau occurs. This band marks a broad altitudinal shift from transitional (< dm/yr) to active (> dm/yr) rock glacier types (Bertone et al., 2024).

.

The multi-operator RoGI exercise resulted in the identification of 39 certain rock glacier units, and 13 uncertain features. The InSAR-based MA indicate velocities ranging from < 1 cm/yr to > 100 cm/yr. The assigned KA has contributed to classify the RGU activity as relict (6 RGU), transitional (19 RGU), and active (14 RGU). The average size of the mapped rock glaciers is ~0.05 km$^2$ based on the extended outlines.

### 4.8 RoGI area 12-1 GL (Disko Island, Greenland)

RoGI area 12-1 is located along the northeastern coast of Disko Island, Greenland (central lat./long. location: 69°51' N, 52°33' W). The area covers an extent of approx. 82 km$^2$. The elevation ranges from peaks up to ~1330 m a.s.l mountain tops in the southwestern part, down to sea level.

There is a high density of rock glaciers in the area, previously explained by the combination of continuous permafrost and the abundance of heavily weathered basaltic bedrock (Humlum, 1996). Previous studies have already pointed out that tongue-shaped rock glaciers fed by glaciers in the hinterland are difficult to distinguish from debris-covered glaciers (Humlum, 1982).

The multi-operator RoGI exercise resulted in the identification of 29 certain rock glacier units, and 19 uncertain features. The InSAR-based MA indicate velocities ranging from < 1 cm/yr to > 100 cm/yr. The assigned KA has contributed to classify the RGU activity as uncertain (2 RGU), relict (1 RGU), relict uncertain (9 RGU), transitional (8 RGU), and active (9 RGU). The averaged size of the mapped rock glaciers is ~0.05 km$^2$ based on the extended outlines.

### 4.9 RoGI area 13-1 KA (Northern Tien Shan, Kazakhstan)

RoGI area 13-1 is located in the central part of Ile Alatau (also Zailiskiy Alatau), Northern Tien Shan in Central Asia (central lat./long. location: 43°0' N, 77°1' W). The area is located in Southern Kazakhstan, close to the border with Kyrgyzstan. The area covers an extent of approx. 59 km$^2$. The elevation ranges from peaks up to ~4365 m a.s.l in the eastern part, down to ~2570 m a.s.l in the valley in the northwest.

Previous research has shown that rock glaciers are abundant in entire northern Tien Shan (Gorbunov and Titkov, 1989; Kääb et al. 2021; Titkov, 1988). More detailed investigations of the rock glaciers in the central part of northern Tien Shan highlighted the existence of several large complex rock glaciers, which originate in elevations where permafrost is very likely and flow down to elevations where permafrost is sporadic (Bolch & Gorbunov, 2014; Marchenko et al. 2001). Many rock glaciers in this region are highly active with average surface velocities of 1 to more than 2.5 m/yr (Gorbunov et al. 1992, Kääb et al. 2021).

The multi-operator RoGI exercise resulted in the identification of 14 certain rock glacier units, and 16 uncertain features. The InSAR-based MA indicate velocities ranging from 1–3 cm/yr to > 100 cm/yr. The assigned KA has contributed to classify the RGU activity as transitional (1 RGU) and active (13 RGU). The averaged size of the mapped rock glaciers is ~0.35 km$^2$ based on the extended outlines.

### 4.10 RoGI area 14-1 US (Brooks Range, U.S.A.)

RoGI area 14-1 is located in the Brooks Range, in Northern Alaska, U.S.A. (central lat./long. location: 68°6' N, 149°58' W). The area covers an extent of approx. 21 km$^2$. Elevation ranges from peaks up to ~2070 m a.s.l in the central part of the area, down to ~1120 m a.s.l. in the valleys further North.

The area is underlain by continuous permafrost. Previous research in this area mapped rock glaciers between 900 and 2000 m a.s.l., and occurring mainly on the north side of the Brooks Range (Calkin, 1987; Ellis and Calkin, 1979; Ikeda et al., 2008). Previous measured rates of two rock glaciers in the 1980s were 10 and 40 cm/yr (Calkin, 1987).

The multi-operator RoGI exercise resulted in the identification of 14 certain rock glacier units, and 14 uncertain features. The InSAR-based MA indicate velocities ranging from 1–3 cm/yr to > 100 cm/yr. The assigned KA has contributed to classify the RGU activity as relict (3 RGU), transitional (2 RGU), and active (9 RGU). The averaged size of the mapped rock glaciers is ~0.07 km$^2$ based on the extended outlines.

### 4.11 RoGI area 15-1 AR (Central Andes, Argentina)

RoGI area 15-1 is located in the Central Andes, West from Mendoza, Argentina (central lat./long. location: 32°59' S, 69.34° W). The area covers an extent of approx. 55 km$^2$. Elevation ranges from up to ~5530 m a.s.l for the southernmost peaks, down to ~3570 m a.s.l in the valley in the northern part of the area.

Previous studies reported an exceptional density of rock glaciers in the Central Andes of Argentina (Zalazar et al., 2020), where permafrost occurs from ~3600 m a.s.l. upwards (Trombotto Liaudat, 2000). Recently, significant surface displacements between 0.37 and 2.61 m/yr were assessed for large complex rock glaciers in the region (Blöthe et al.,2020), and short-term active layer monitoring documented the degradation of ice-rich permafrost in rock glaciers (Trombotto Liadat and Bottegal, 2019).

The multi-operator RoGI exercise resulted in the identification of 70 certain rock glacier units, and 18 uncertain features. The InSAR-based MA indicate velocities ranging from 1–3 cm/yr to > 100 cm/yr. The assigned KA has contributed to classify the RGU activity as relict uncertain (3 RGU), transitional (19 RGU), active (42 RGU), and active uncertain (6 RGU). The averaged size of the mapped rock glaciers is ~0.12 km$^2$ based on the extended outlines.

### 4.12 RoGI area 16-1 NZ (Southern Alps, New Zealand)

RoGI area 16-1 is located in the Ben Ohau Range, part of the Southern Alps of New Zealand (central lat./long. location: 43°59' S, 170°3' E). The study area covers an extent of approx. 7 km$^2$. Elevation ranges from peaks up to 2431 m a.s.l for the highest peak in the north, down to ~1600 m a.s.l in the westernmost valley.

In two previous studies in the study area, Sattler et al. (2016) identified two relict, four inactive, and six active rock glaciers, based on aerial image analysis only, while Lambiel et al. (2023) reported the presence of ten transitional and two active rock glaciers, using Sentinel-1 InSAR data.

The multi-operator RoGI exercise resulted in the identification of 24 certain rock glacier units, and 6 uncertain features. The InSAR-based MA indicate velocities ranging from 1–3 cm/yr to > 10–30 cm/yr. The assigned KA has contributed to classify the RGU activity as relict (9 RGU), transitional (10 RGU), and active (5 RGU). The averaged size of the mapped rock glaciers is ~0.03 km$^2$ based on the extended outlines.

 **4.13  Results summary across all areas**

In total, 337 "certain" rock glaciers were identified and characterised, and 222 additional landforms were identified as "uncertain" (Figure 5). The level of uncertainty varies and reflects the geomorphological complexity of each area. On average, about 40% of the landforms remain "uncertain". At these locations, the inventorying teams judged that we need more precise data and/or field visits to finalise the assessment.

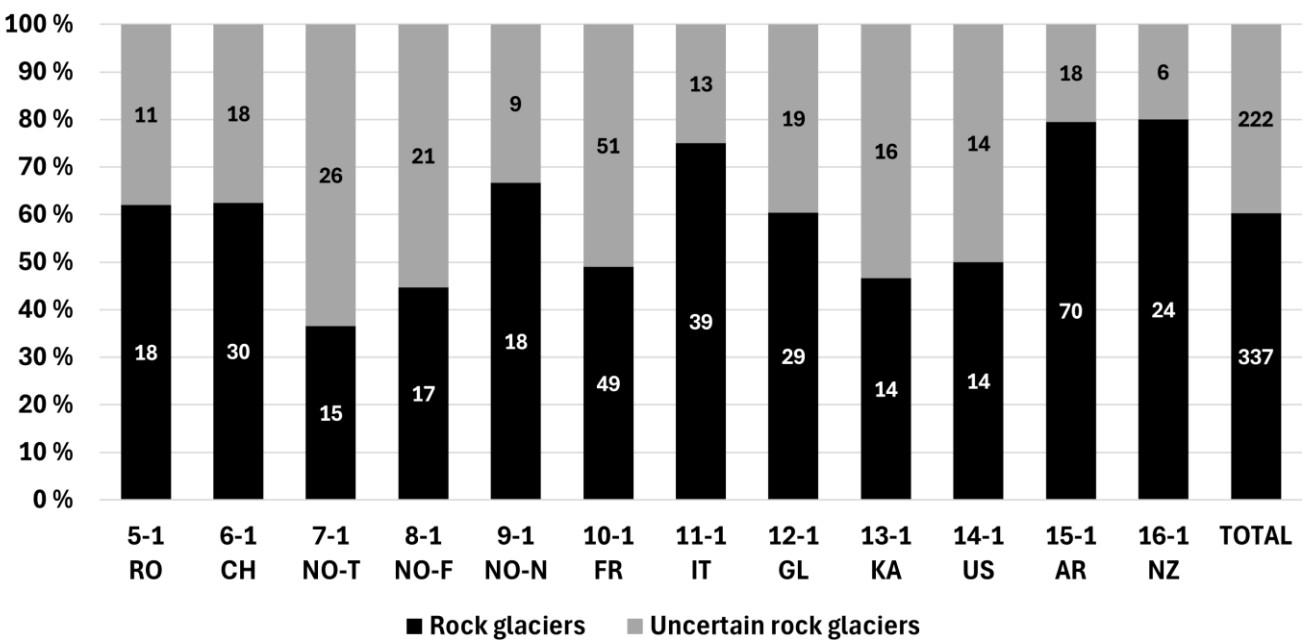

**Figure 5: Relative distribution of the Rock Glacier Units (RGU) identified as "certain" (black) or "uncertain" (grey) in each RoGI area resulting from the consensus-based final Primary Marker (PM) layers. The numbers written in the bars correspond to the absolute numbers of landforms. The area numbers and the acronyms of the corresponding countries are used as x-axis legend (RO: Romania, CH: Switzerland, NO: Norway (NO-T: Troms, NO-F: Finnmark, NO-N: Nordenskiöld Land), FR: France, IT:**
**Italy, GL: Greenland, KA: Kazakhstan; US: U.S.A.; AR: Argentina, NZ: New Zealand), according to Table 1 naming convention. Further analysis in the second phase of the exercise (outlining and characterisation of the attributes) was performed on the "certain" rock glaciers only.**

The InSAR-based MA polygons have a wide range of velocities, both between and within the areas (Figure 6). The MA layers were used to assign the kinematic attribute (KA) of each RGU, which then was used to assess the activity (Figure 7).

The kinematic and activity attributes of the PM files are therefore related to the MA layers, but the respective information is also complementary. While the activity is a convenient way to summary the rock glacier state, the MA layers provide a more comprehensive overview of the distribution of the rock glacier creep rate. There are overall more MA polygons than RGU PM due to spatial heterogeneities in velocity (i.e., several MA over the same RGU).

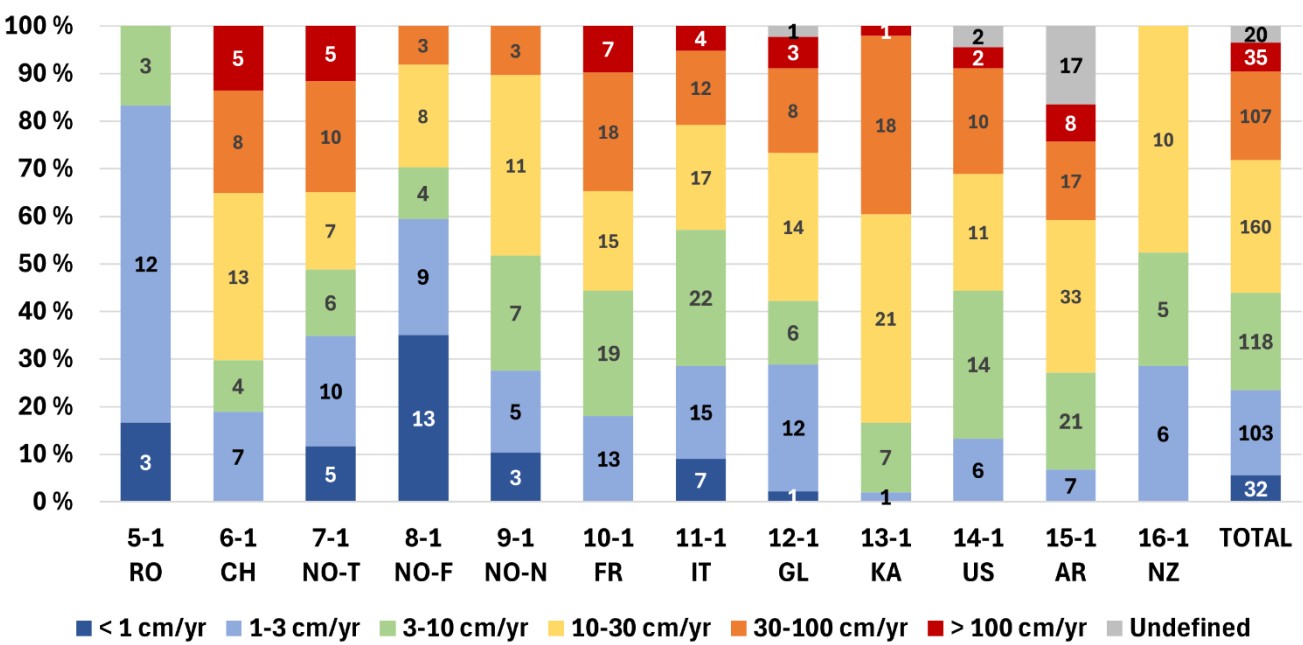

**Figure 6: Relative distribution of the velocity classes of the InSAR-based Moving Areas (MA) in each RoGI area resulting from consensus-based final MA layers. The numbers written in the bars correspond to the absolute numbers of landforms. The area numbers and the acronyms of the corresponding countries are similar to Figure 5 and according to Table 1 naming convention.**

Rouyet et al. ESSD revised manuscript 19/05/2025

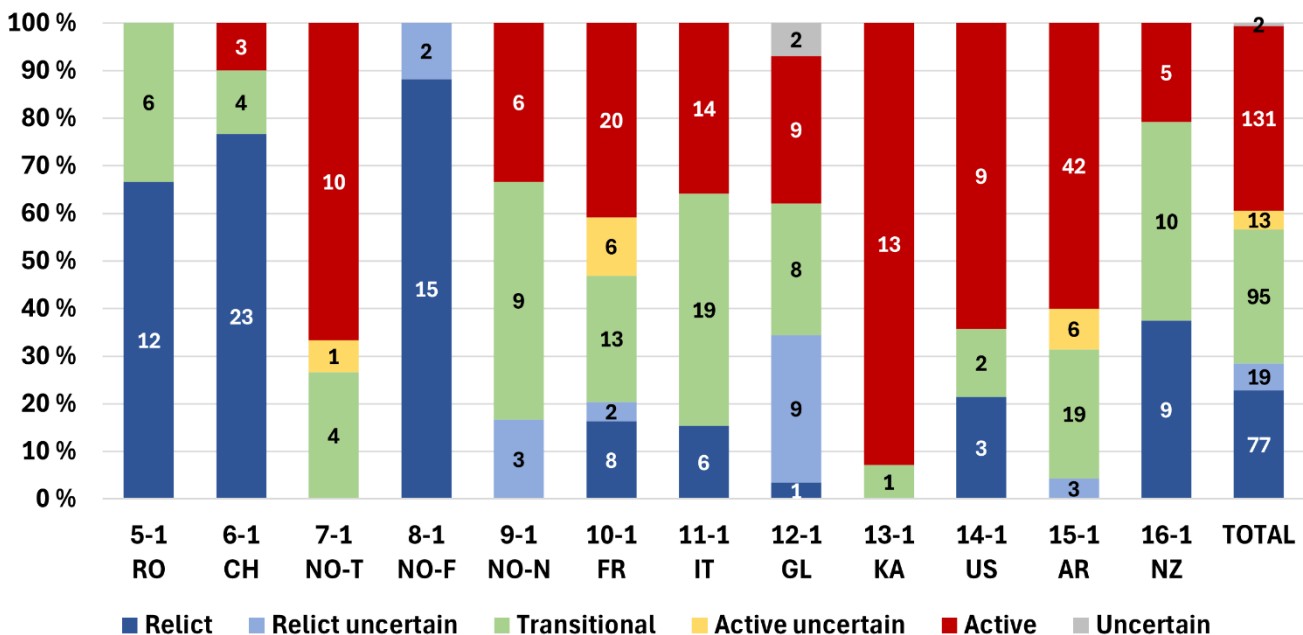

**Figure 7: Relative distribution of the RGU activity (active, active uncertain, transitional, transitional uncertain, relict, relict uncertain, uncertain), documented as attribute in the consensus-based final Primary Marker (PM) layers. The numbers written in the bars correspond to the absolute numbers of landforms. The area numbers and the acronyms of the corresponding countries are similar to Figure 5 and according to Table 1 naming convention.**

Based on the extended outlines, the RGU have a typical size ranging between 0.01 and 0.25 km$^2$ (median value of each area, Figure 8). The boxplots indicate large differences in size between and within the areas. It should be noted that in areas dominated by large rock glaciers (e.g., area 12-1 GL; area 13-1 KA), small talus-connected rock glaciers may have been overlooked. The size of the areas significantly varies (ranging from 7 to 82 km$^2$, see Table 1). The size of the mapped landforms, as well as the number of certain and uncertain RGU, in respect to the size of the area, are also highly variable (Figure 9). Some areas are characterised by many small landforms (e.g., area 6-1 CH; area 16-1 NZ), while others are dominated by few large rock glacier units (e.g., area 12-1 GL; area 13-1 KA)

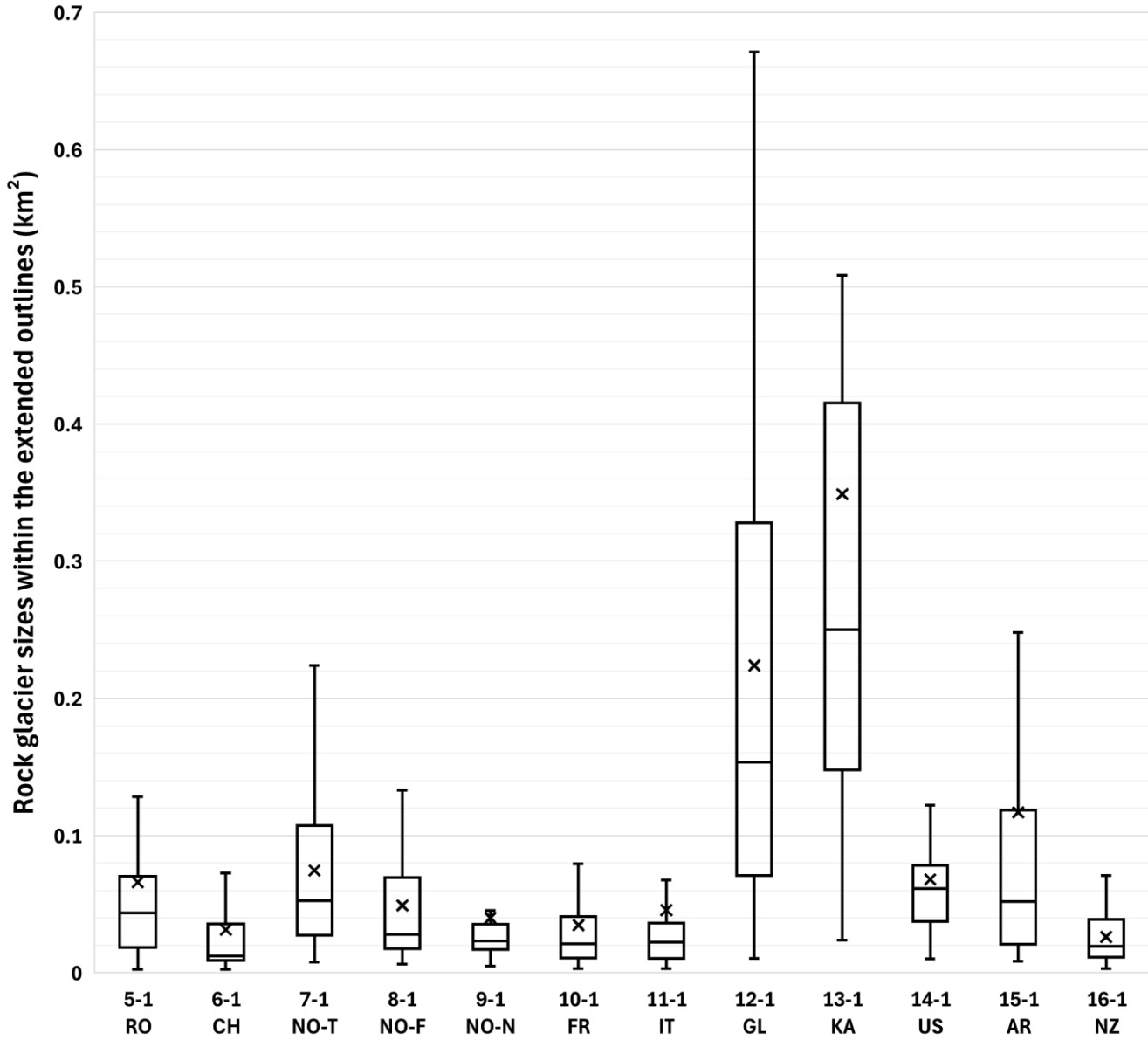

**Figure 8: Range of RGU sizes within the extended Geomorphological Outlines (GO) in each RoGI area, resulting from the consensus-based final GO layers. The horizontal lines in the boxes indicate the median values. The lower and upper limits of the boxes indicate the 1st and 3rd quantiles. The whiskers highlight the maximum and minimum values. The crosses indicate the averaged sizes. The area numbers and the acronyms of the corresponding countries are similar to Figure 5 and according to Table**

**1 naming convention.**

Rouyet et al. ESSD revised manuscript 19/05/2025

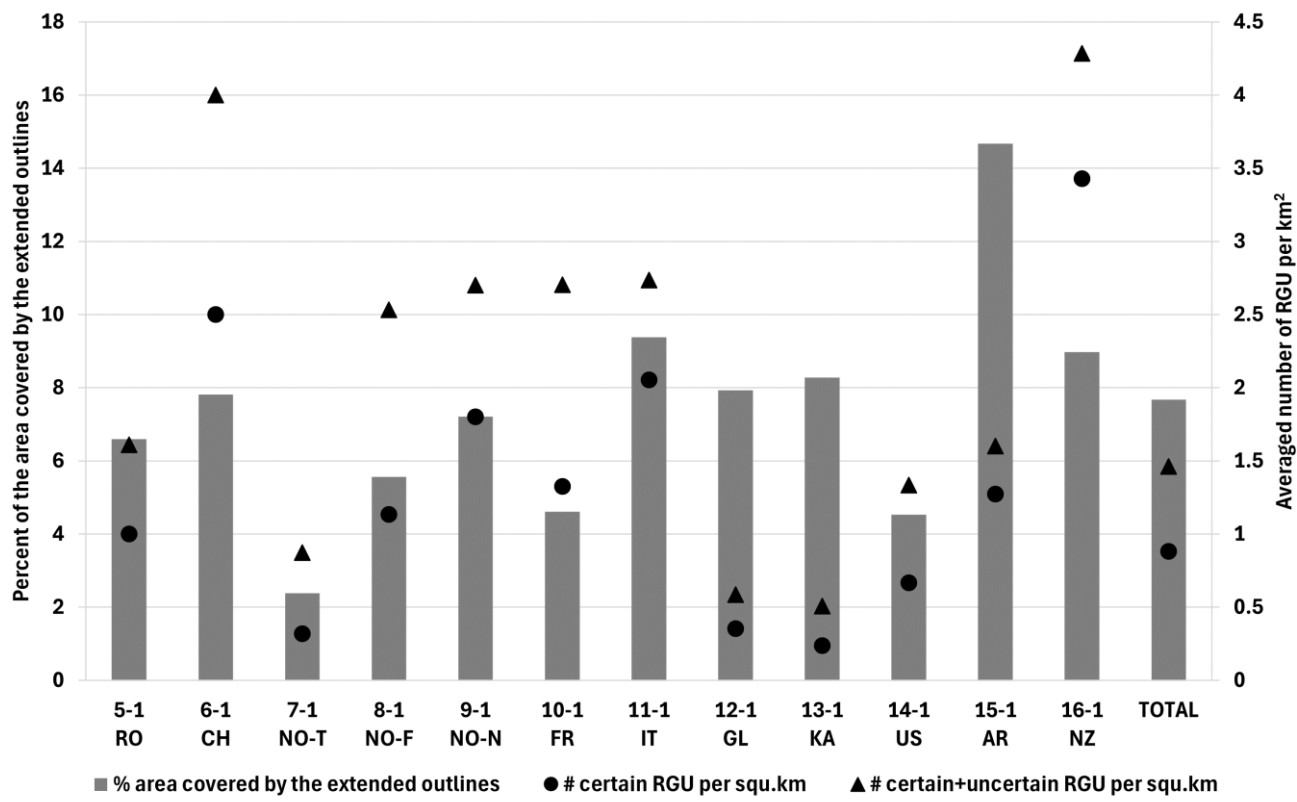

**Figure 9: Density of rock glaciers in the studied areas. The grey bars show the percent of the area covered by rock glaciers, according to the mapped extended outlines. The corresponding values are shown on the primary vertical axis on the left. The black symbols (dots: certain rock glaciers; triangles: certain and uncertain rock glaciers) show the numbers of identified RGU in respect to the size of the area (number per km². The corresponding values are shown on the secondary vertical axis on the right. The area numbers and the acronyms of the corresponding countries are similar to Figure 5 and according to Table 1 naming convention.**

## 5 Uncertainties and limitations

### 5.1 Documentation of uncertainties and limitations

In the attribute tables of the three GeoPackage files, various fields document the reliability of the mapping and morpho-kinematic assessment, according to identified uncertainties and limitations:

- For the PM files, an attribute "uncertain" describes ambiguous areas that should be investigated in the future (need for additional data and/or field visit). For educational purposes, an attribute "not a rock glacier" could also be used to highlight landforms that are likely to be misinterpreted as rock glaciers. The level of uncertainty and complexity can be highlighted for many morpho-kinematic attributes, either in the selectable categories (for example "active uncertain", "transitional uncertain", and "relict uncertain" for the attribute "Activity") or using an additional reliability attribute for

the kinematic assessment (Appendix A). Additional comments describing the uncertainty sources and ambiguities in the interpretation can be written in two "Comment" fields.

- For the MA files, the reliability (or the degree of confidence) of the results is qualitatively documented in accordance with the quality of the detection, the delineation of the Moving Areas based on the available InSAR data, the signal interpretation and the resulting velocity estimation (Appendix B). When medium–low reliability is set (uncertain InSAR signal and/or unclear MA outlines), information on the uncertainty sources and ambiguities in the interpretation can be described in a "Comment" field.

- For the GO files, the reliability of the delineation at different locations of the rock glacier (front, left/right lateral margins, upslope boundary) is estimated with a score of 0 (low), 1 (medium), or 2 (high). It consists of a qualitative assessment depending on the data quality and the geomorphology complexity of the landform (Appendix C). The automatic summation of the scores (0–8) gives a general estimate of the outline reliability for the entire landform. Information regarding the data source(s) used for the delineation and the uncertainties impacting the reliability of the resulting polygon can be documented in a "Comment" field.

## 5.2 Summary of uncertainties and limitations for each RoGI product

Here we summarise the observations about the uncertainties and limitations of the three output files, based on the results in the 12 areas and the feedback of the operator teams. Most challenges are common for all areas, while a few are affecting specific areas only. The main identified uncertainties and limitations of each output product are described in the following sections and summarised in Table 3. Despite the effort to standardise the procedure and reduce the differences between the areas, we acknowledge that discrepancies remain in the final products. These are due to the different levels of geomorphological complexity, the variable numbers of landforms and the density of their distribution, as well as the heterogenous data quality, local knowledge and research history. In case of operator discrepancies, the decisions were taken at the team level, ensuring homogeneity within each area. Major questions were discussed during PI coordination meetings and communicated across the teams thanks to operators working in several areas. The parallel timeline of the work in all areas contributed to a good communication on the common challenges, but did not discard all risks of inter-regional differences and subjective treatment.

### 5.2.1 Uncertainties and limitations of the PM products

- The quality of the PM products depends on the availability, resolution, and quality of the source data, which varies among the areas. Optical imagery affected by shadows, clouds, or snow cover led to increased uncertainty in some areas. Warm regions in the marginal permafrost zone can be dominated by relict landforms with dense vegetation cover that may hinder detailed mapping based on passive optical remote sensing only. In such areas, terrain hillshades from high-resolution LiDAR DEM are highly valuable as complementary input data. The availability and quality of such products are however variable from a region to another. In areas dominated by large glacier-connected or glacier-

forefield-connected rock glaciers, small talus-connected rock glaciers tended to be overlooked, which may explain the different size distribution between the areas (see Figure 8). The identification of small and relatively shallow rock glaciers developed on debris-mantled slopes was challenging, as they often do not exhibit well-defined rooting zones and lateral margins.

- InSAR was useful for detecting rock glaciers that may have been missed when only looking at optical images. However, it also added an additional source of variability between the regions, because the data availability and properties vary from a region to another (Table 2). In areas with multi-temporal PS/DS InSAR data, the detection capability to low velocity was increased. In areas with X-band SAR data, interferograms with higher spatial resolution were provided, which allowed for detecting smaller moving landforms. In areas with L-band SAR data and 6-days Sentinel-1 repeat-pass, the maximal detection capability was increased.

- The level of geomorphological complexity and the interactions with other glacial and periglacial processes vary among the areas. Several teams reported the challenges of discriminating landforms due to the glacier–rock glacier continuum or between rock glacier landforms and complex morainic systems in the case of rock glaciers derived from former (late-glacial) glacier-forefield. The imbrication with other types of periglacial processes also leads to ambiguities in the landform discrimination (e.g., large coarse solifluction lobes and rockslide deposits). In such cases, the final products include many "uncertain" PM and several cases with unclear upslope connections. In areas with landslides, there were difficulties in assigning the type of upslope connection. The "landslide-connected" upslope unit is somewhat ambiguous, as it often practically means that there is a poly-connection (talus+landslide). In such cases, the high level of discrepancy between operators required some discussions to agree on a final category.

- One challenge reported by all teams is related to the variable level of details applied to discriminate landforms with complex morphology. There were discrepancies among the operators in the way to interpret multi-unit systems and discriminate the units. In complex cases, some operators considered the landforms as one main complex unit (one PM), while others identified several units (several PM). After discussions, consistent solutions were found within each area. There are remaining differences between the areas due to the variable geomorphological complexity.

- The quality of the attribute characterisation depends on the complexity of the study area. In cold-climate regions with continuous permafrost, one challenge is related to the kinematic and activity attributes. Although the landforms are "active" in the traditional sense (i.e., intact, with presence of permafrost), some are very slowly creeping and so fall into the transitional or relict category according to the current RGIK definition (low efficiency of sediment conveyance). The activity is also challenging to assess in areas where the contrast in surface material between the rock glacier surface and the front is generally low. In such cases, the front is generally smooth, which makes it is hard to discriminate active from transitional rock glaciers without kinematic information.

### 5.2.2 Uncertainties and limitations of the MA products

- The accuracy of the kinematic analysis varied between areas due to unequal data availability. Some areas had a variety of InSAR data from different SAR sensors and processed with different algorithms (single interferograms, stacking and PSI), providing a wide range of detection capabilities. In such cases, the areas with no MA could reliably be interpreted as "no movement" (i.e., movement < 1 cm/yr), and high to very high velocity can be discriminated (30–100 cm/yr or > 100 cm/yr). However, some areas had fewer datasets and longer Sentinel-1 repeat-pass time interval (extra-European
areas), which led to reduced maximal detection capability. Using Sentinel-1 12 days repeat-pass only, it is not possible to discriminate 30–100 cm/yr and > 100 cm/yr, which leads to discrepancies in the way to assign the m/yr and > m/yr categories (kinematic attribute in the PM layer).

- The InSAR interpretation resulted in discrepancies between operators. All teams reported that it was the hardest step of the RoGI procedure due to variable backgrounds of the operators. A consensus-based process was difficult to perform,
due to variable levels of experience with InSAR, the different ways to look at all available datasets, and the variable levels of details in systematically outlining the MAs. In most cases, the same MA were similarly identified and delineated, but the velocity classes were sometimes assigned differently. Despite this challenge, many operators reported that the work was highly educative, and all teams had at least one operator well-experienced with InSAR, which ensured high quality in the final results.

- In general, major fast-moving MA were detected with few variabilities among operators, while small and slow MA were more difficult to interpret. In marginal permafrost zones, there is a dominance of transitional and relict rock glaciers characterised by little (or no) movement. The MA are small and with low velocity, and therefore hard to identify compared to other areas characterised by strong InSAR signal on interferograms with short time intervals between the compared images. The documentation of low velocity MA requires the availability of multi-temporal
PS/DS data with other properties and interpretation constraints than single interferograms.

- A general challenge with InSAR analysis is to ensure that the detected movement is representative of the rock glacier creep rate and not significantly affected by other processes (e.g., landslide, solifluction, thaw subsidence). Analysing a diverse set of interferograms (various SAR geometries, time intervals, months and years) allow to reduce the risk of misinterpretation by providing complementary information about the spatial and temporal characteristics of the
movement pattern. However, we cannot fully discard the possibility that some MA identified on a rock glacier might be affected by other processes. In case of rapid permafrost degradation, the detected movement may correspond to a mixed signal from downslope creep and subsidence due to ice core melt. In cold regions with continuous permafrost, the ground is highly dynamic during the thawing season, which makes it difficult to dissociate the InSAR signal on the rock glacier from surroundings areas that also move. When analysing small and slowly creeping talus-connected rock
glaciers, it was sometimes challenging to discriminate the movement associated with rock glacier creep from other processes, such as thaw subsidence in ice-rich lowlands located directly at the foot of the mountain ridges.

Rouyet et al. ESSD revised manuscript 19/05/2025

### 5.2.3    Uncertainties and limitations of the GO products

- The main difficulty was to delineate the upper boundary between the rock glacier and its contributing area, depending on the type of upslope connection. For glacier-connected or glacier-forefield-connected rock glaciers, the location of the upper boundary was often ambiguous, and the corresponding outline reliability therefore set to "low" in the attribute table. For talus-connected rock glaciers, there were discrepancies on how to draw the upper outline: straight line on the upslope area of the rock glacier, versus a curved connection to avoid the inclusion of talus cones feeding the rock glacier. The teams discussed this challenge and agreed on drawing a curved line, continuing the delineation of the front and margins while following the topography. When the location of the boundary was uncertain, the upper outline reliability was set to "low" or "medium" in the attribute table.

- In some cases, the delineation of the front was challenging, especially if the toe of the rock glacier was reworked by other processes, such as solifluction. Smooth fronts and rounded ridges and furrows, often associated with relict and transitional landforms, may lead to ambiguous delineation of the restricted outlines. For a rock glacier developing on a steep slope, the front may also be difficult to distinguish. Some problems were for instance identified in cases of exaggerated fronts blended with the downside talus slope. Small rock glaciers, such as debris-mantled-connected rock glaciers, or embryonic talus-connected landforms (protalus ramparts) often had ambiguous lateral margins, challenging for outlining. Such complicated cases were discussed during team meetings to find a mutually agreeable solution. When the location of the boundary was uncertain, the front and/or lateral outline reliability was set to "low" or "medium" in the attribute table.

- Complex rock glacier systems with several units were the most challenging landforms to outline. The delineation was especially difficult in the case of adjacent landforms or several generations of partly overlapping rock glacier units. In some cases, several units initially identified with different PM in the first phase of the exercise were not outlined separately in the second phase (too complex). The two phases (PM identification and GO delineation) were performed iteratively. When the outlining process highlighted that multi-unit discrimination was bringing too much uncertainty, the PM numbers and locations were revised (simplification).

- Combining different data sources with variable acquisition times, snow/vegetation covers, and sunlight directions helped interpreting and mapping some rock glaciers. On the other hand, some areas are affected by georeferencing shifts between the different optical data sources available in the online services. These shifts may explain some discrepancies among operators, depending on the main source used to digitalise the boundaries. The scale and level of details used to perform the outlining work also varied between the operators. This challenge did not affect the consistency and quality of the final products that were discussed within the teams and accordingly revised by each PI. The data source used for the final outlines and the time it applies is specified in the attribute table. The results apply for the period during which the outlines were drawn. If viewing the results with a more recent background imagery, new shifts may occur due to the regular updates of the WMS data sources and their variable quality.

**Table 3: Overview of the main uncertainties and limitations of the RoGI products and how they apply to the 12 areas. The crosses (X) show where the problem has been explicitly reported by the RoGI team/PI. The circles (O) show where the problem might happen for specific landforms, but had not been reported has a main limitation by the RoGI team/PI. The area numbers are similar to Figure 5 and according to Table 1 naming convention.**

| | 5-1 | 6-1 | 7-1 | 8-1 | 9-1 | 10-1 | 11-1 | 12-1 | 13-1 | 14-1 | 15-1 | 16-1 |
|---|---|---|---|---|---|---|---|---|---|---|---|---|
| **PM detection and characterisation** | | | | | | | | | | | | |
| Optical imagery affected by shadows, clouds, snow | O | O | O | O | X | O | O | X | O | X | O | O |
| Dense vegetation cover on relict rock glaciers | X | | | | | | | | | | | |
| Dominance of large RGU and small RGU likely overlooked | | | | | | | | O | X | X | X | |
| Ambiguous imbrication of periglacial landforms | O | O | X | X | O | X | O | O | O | X | O | |
| Ambiguous rock glacier and glacier/forefield continuum | O | X | O | O | O | O | O | O | X | X | X | |
| Variable categorisation of landslide-connected RGU | O | O | X | O | O | X | O | O | O | O | O | |
| Difficulty to select RGU for complex multi-unit systems | O | O | X | O | O | O | O | O | X | X | X | O |
| Ambiguity in activity in Arctic cold regions with slow/no MA | | | | | X | O | | | O | | | |
| Difficulty to discriminate active/transitional | | O | O | | O | O | O | O | O | O | O | X |
| **MA detection, delineation and characterisation** | | | | | | | | | | | | |
| Fewer available Sentinel-1 images and longer repeat-pass | | | | | | | | X | X | X | X | X |
| Challenge of velocity estimate for operators with little InSAR experience | X | X | X | X | X | X | X | X | X | X | X | X |
| Difficulty to document and interpret small and slow MA | X | O | O | X | O | O | O | O | O | O | O | O |
| Difficulty to discriminate creep from other processes | O | O | O | X | X | O | O | X | O | X | O | O |
| **GO delineation and characterisation** | | | | | | | | | | | | |
| Uncertainty in the delineation of the upper boundaries | X | X | X | X | X | X | X | X | X | X | X | O |
| Uncertainty in the delineation of eroded, reworked or exaggerated fronts | X | X | O | X | O | X | X | O | O | X | X | O |
| Unclear lateral margins for small rock | O | O | O | X | O | O | X | O | O | O | O | O |

| glaciers | | | | | | | | | | | | |
|---|---|---|---|---|---|---|---|---|---|---|---|---|
| Difficulty to outline complex RGS with multiple RGU | O | O | X | O | O | O | O | O | X | X | X | O |
| Variable quality of optical imagery and georeferencing shifts | O | O | O | X | X | O | O | X | X | X | O | O |

## 5.3 Quality assessment of the multi-operator RoGI procedure

Here we summarise the observations about the multi-operator RoGI procedure, based on the results in the 12 areas and the feedback of the operator teams.

### 5.3.1 Value of the RoGI exercise and the multi-operator procedure

- The steps and instructions of the exercise were generally assessed as clear and easy to follow. The operators reported that they liked the structure and clarity of the provided GIS and data packages. Thus, it is promising to apply the same structure in new regions and therefore ensure consistency in future RoGI data compilation.

- Each team had two multi-operator meetings, with 3–10 people attending. The size of the teams proved ideal for such an exercise, as more people would have been challenging to manage and ensure efficient discussions. In some cases, the digital meetings were complemented with email interactions (e.g., sharing of comments in documents, prints screens, powerpoints, and sending recording of meetings). All types of communication were found valuable, both for personal learning purpose and for improving the quality of the final products.

- Having operators with different skills and backgrounds was found to bring in added value to the final results. The combination of different points of view and experiences from several regions around the world ensured that various morpho-kinematic elements were identified and taken into consideration.

- Although InSAR interpretation has been identified as the most challenging step due to little experience for some operators, several teams report that the data were valuable at different levels. InSAR was useful for identifying moving rock glaciers, in addition to providing a semi-quantitative information about their creep rates. When used iteratively with PM detection (Figure 2), the MA step allowed for including landforms that would have been overlooked based on geomorphological criteria only. It was especially valuable in areas where optical imagery was affected by shadows, clouds, or snow cover.

### 5.3.2 Challenges and suggestions to improve the RGIK procedure and guidelines

- The consensus-based procedure generally worked well for the PM identification, the GO delineation, and the categorisation of key attributes (e.g., upslope connection, kinematics, and activity). However, some steps cannot be comprehensively assessed during team meetings. It is for example not feasible to collectively discuss all details of the

InSAR interpretation. Practically, the PIs compared their own results with those of the other operators, and corrected their results when mistakes were found. A comprehensive consensus-based process can work but only on a small set of rock glaciers, which could then be used for adjusting the assessment criteria before upscaling.

- The InSAR interpretation was challenging for operators without past experience with these types of data. Despite discrepancies in the quality of some individual results, that issue did not impact the final products, as each team included at least one person with InSAR experience. Nevertheless, the teams suggested various ways to improve this part in the future, such as 1) adding new examples in the guidelines on how to read the interferograms, 2) splitting the multi-operator process into two separate teams (one with InSAR expertise focusing on the MA part, one with geomorphological expertise focusing on the PM/GO and using the final MA for the kinematic assessment), 3) pre-processing the data and providing the velocity products in formats that are easier to interpret by non-experts.

- The assignment of the activity attribute based on geomorphological and/or kinematic criteria requires clarification in the guidelines. The InSAR analysis led to the generation of a MA layer with polygons highlighting where movement has been detected. For characterising the kinematics and the activity, the operators used the MA layer as input. However, some rock glaciers are not covered by any MA. In such cases, it was recommended to avoid overinterpreting the absence of detected movement, because a rock glacier without any MA may mean two different things: 1) there is no movement or too low to be detected, 2) the data quality and/or coverage did not allow for detecting it. In such cases, some operators only focused on geomorphological criteria to assign the activity. A kinematic attribute with low velocity and low reliability index may have been documented, but was not used to set the activity. For other operators, the lack of movement evidence has been used in synergy with geomorphological evidence, as an additional indicator confirming the geomorphological interpretation.

- As part of the GO step, the upper outline between rock glacier and its contributing area was identified as the most challenging part to delineate. The way to draw the upper outline when there is a high level of uncertainty could be improved in the RGIK guidelines, based on additional examples, for different landform types and from different places around the world. The scale of digitalisation was not specified at the beginning of the exercise, which led to discrepancies in the outlining level of details and size of the considered landforms. The way to document the data source, imagery date and scale of analysis could be improved. When using Bing, Google or ESRI WMS imagery, it is important to specify the main data source, and the date the work has been performed as these open services have differences and frequent updates, which mean lead to GO shifts. It was encouraged to do it in the field "Comments" or "Kin.Comments" but other elements could be written in these fields, which led to variable metadata documentation depending on the operator.

- Several operators commented that there were many variables to document. The entire inventory process was consequently time-consuming, which led to variable levels of details. It should be noted that in the framework of this exercise, all steps were required although several elements are presented as "optional" in the guidelines. For example, the GO are valuable but are not mandatory to draw. A combination of PM with and without GO is possible within the

same RoGI. One could decide to delineate a large system and mark the locations of several units using PM only (i.e., without outlining at the same level). More compact versions of the RoGI protocol could be developed to avoid discouraging some groups to follow it. Alternative ways to summarise the essential information contained in the RGIK guidelines (short check-list document, flow-chart with link to necessary definitions, video tutorial, etc.) may also help RoGI operators to have a quick overview on the main tasks to perform.

Overall, despite discrepancies in the individual results, the above issues did not impact the final products. Consistent solutions were found after discussion within and among the team(s).

## 6 Conclusion: potential use and applications

The multi-operator RoGI exercise performed in 2023 involved 41 people who applied the RGIK guidelines in 12 areas spread around the world. This unique international initiative fulfilled the four initial objectives outlined at the end of Section 1. First, we demonstrated that it was feasible to apply common RoGI guidelines and procedure in various mountainous environments. All teams acknowledged that the initiative was highly instructive, thanks to the lively discussions in team meetings, the diversity of backgrounds and experiences, and the possibility to perform the work in various geomorphological contexts. Second, we identified various limitations (see Section 5) that will serve to improve the RGIK guidelines in the future. Third, we developed standardised GIS templates for homogenizing the production of future RoGI and providing training tools for the community. The GIS templates and two online exercises are already available on the RGIK website. Fourth, we compiled and disseminated a homogenised set of RoGI in 12 diverse regions.

The resulting dataset has the potential to be used for several applications. Here we discuss four potential uses:

- **Further investigation in the selected areas and RoGI upscaling**: The exercise was performed on relatively small areas (7–82 km$^2$) to make it feasible to apply the demanding procedure described in Section 2. All PIs and involved research groups acknowledged the educational value of the process and that the lessons learnt during the exercise will contribute to continue their work in the regions, with the long-term objective to upscale the RoGI to entire mountain ranges. The landforms, for which the current characterisation was uncertain with the applicable data, may be investigated further during future targeted fieldwork campaigns or when new datasets become available.

- **Rock glacier selection for Rock Glacier Velocity (RGV) monitoring:** Following the recent acceptance of RGV as a new parameter of the ECV Permafrost (Hu et al., 2025; Streletskiy et al., 2021; WMO, 2022), one important task of the community is to monitor the interannual velocity changes of selected rock glaciers, using in situ and/or remote sensing techniques. It is highly recommended to have a good understanding of the rock glaciers selected for long-term monitoring and exploitations as climate change indictor. The development of comprehensive RoGI in several regions is therefore a valuable first step to design monitoring strategies in each area (RGIK, 2023c).

- **Educational training tools for enhancing the systematic generation of RoGI worldwide:** The international multi-operator exercise highlighted the variety of rock glacier morphologies and characteristics across the selected mountain ranges, showing the importance of illustrating the RGIK guidelines with examples from different regions. The operator comments show the need to promote the guidelines with alternative tools (e.g., compact version of the RoGI protocol, short check-list document, flow-chart summarizing the main steps with links to necessary definitions, video tutorial, additional GIS training tools based on the present dataset, etc.). New training material, partly based on our RoGI dataset, may contribute to promoting and supporting the generation of RoGI in under-studied regions.

- **Training data for automated inventorying techniques:** There is a growing interest in the community for developing automated solutions for RoGI generation at a large scale, using machine learning (Erharter et al., 2022; Mahanta et al., 2024; Robson et al., 2020; Sun et al., 2024). Per definition, machine learning requires high-quality datasets to train the model. Transferability is typically a challenge. If the input data is clustered in a small area, the model may fail to map rock glaciers in another region with different conditions. In this respect, despite the few landforms in each area, our dataset covers a wide range of topographic, geological, and climatic conditions. To our knowledge, this is the first publicly released dataset that combines RoGI in ten different countries and five continents, which will hopefully be valuable for machine learning applications.

**Data availability**

The final PM/MA/GO dataset is available on Zenodo (Rouyet et al., 2025; https://doi.org/10.5281/zenodo.14501398). The GeoPackage (gpkg) templates for performing similar RoGI in other areas, and exercises based on the QGIS tool are available on the RGIK website (https://www.rgik.org). The data can also be viewed in a dedicated WebGIS tool (https://bigweb.unifr.ch/Science/Geosciences/Geomorphology/Pub/Website/CCI/CCI_qgis2web_2025_04).

**Author contributions**

The ESA CCI Permafrost project is managed by TS. LRo and CP coordinate the mountain permafrost component of the project. The multi-operator exercise was designed by RD, TE and LRo. The RoGI instructions and GIS generic tool were prepared by TE, RD, CP and LRo. The InSAR datasets were processed by TS and LRo. LRo set the timeline of the exercise, led coordination meetings with the PIs and oversaw the work between the areas. LS assisted the coordination of the PI meetings. The multi-operator work in each area was coordinated by the PIs: FS (area 5-1), TE (area 6-1), LRo (area 7-1, 8-1, 9-1), DC (area 10-1), FB (area 11-1), RC (area 12-1), TB (area 13-1), MD (area 14-1), LRu (area 15-1) and CL (area 16-1). RD performed the work in all the areas and contributed to coordinate key decisions across the teams. The PIs finalised the products for their area. Technical correction and final data package compilation was performed by TE, LRo, LS and RD.

LRo led the work on writing and revising the manuscript. Figures and tables were made by TE and LRo. All authors contributed to the final version of the paper.

## Completing interests

The contact author has declared that none of the authors have any competing interests.

## Acknowledgement

We acknowledge the effort of the other operators of the RoGI exercise, listed here in alphabetic order: BODIN Xavier, BRENCHER George, CARRASCO Javiera, COSTANTINI Daniel, DUVANEL Thibaut, FALASCHI Daniel, FARÍAS-BARAHONA David, FERRI Lidia, HANDWERGER Alexander, JOHNS Paula, JONES Nina, KELKAR Kaytan, ONACA Alexandru, PASQUALI Silvia, PECKER Ivanna, POPESCU Razvan, ROBSON Benjamin, SAN MARTIN Cristina, SCHAFFER Nicole, SCHNITZER Monika, SCOTTI Riccardo, SUN Zhangyu, TONIDANDEL David, WEE Julie, WEHBE Mishelle, WENDT Lotte, WOOD Ella, ZALAZAR Laura. We acknowledge the work of Alina Milceva, who helped revising the outlines and setting up the WebGIS tool. We thank Tim Kerr for his open comment, and the two anonymous reviewers for their great referee reports, which provided constructive suggestions to improve the dataset and associated paper.

## Financial support

The initiative is funded by the European Space Agency Permafrost Climate Change Initiative (ESA CCI Permafrost, contract 4000123681/18/I-NB). The work of the Rock Glacier Inventories and Kinematics (RGIK) community has been supported by the International Permafrost Association (IPA), GCOS Switzerland, and SwissUniversities.

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

**Appendix A: Attribute Table of the Primary Marker (PM) files**

Attributes of the Primary Marker (PM) gpkg files (M: mandatory attribute; O: optional attribute). The table only includes the essential definitions necessary to understand the overall meaning of the attributes. The last column refers to the sections of the RGIK guidelines documenting the detailed recommendations for identifying rock glaciers and assigning values to each attribute. At the time of the exercise, the table referred to the sections of several dedicated documents (RGIK, 2022a; 2022b; 2022c). The RGIK RoGI guidelines have since been merged into one reference document (RGIK, 2023a). The following table has therefore been updated accordingly.

| Attribute | Description | Values | RGIK guidelines |
|---|---|---|---|
| fid (M) | Unique identifier of the Primary Marker. | Automatic filling | |
| Landform (M) | 'Rock glacier' is the default value.<br><br>'Uncertain rock glacier': Ambiguous landforms that should be investigated in the future (need for additional data and/or field visit). It provides an option to document the location of suspected rock glaciers that remain uncertain based on the currently available data.<br><br>'Not a rock glacier': This attribute allows the operators to highlight landforms that are likely to be misinterpreted as rock glaciers, for educational purpose. | 0. Uncertain rock glacier<br>1. Rock glacier<br>2. Not a rock glacier | RoGI guidelines chap. 3 (sections 3.1 and 3.7)<br>RoGI guidelines chap. 5 (section 5.1) |
| WorkingID (O) | Practical identifier chosen by the operator (e.g., TYR001, TYR002, ... for an inventory in Tyrol). | Text | |
| Lat. (M) | Latitude of the Primary Marker in decimal degrees. | Automatic filling | |
| Long. (M) | Longitude of the Primary Marker in decimal degrees. | Automatic filling | |
| PrimaryID (M) | RGU + 12 to 15 digits depending on the "Lat.", "Long." values. Always 4 digits after the degrees.<br>(e.g., RGU34567S123456E means 3,4567° South and 12,3456° East) | Automatic filling | RoGI guidelines chap. 5 (section 5.2) |
| Alter.ID1 (O) | Alternative local or regional name | Text | |
| Alter.ID2 (O) | Identifier used in a previous inventory. | Text | |
| Assoc.RGS (O) | Defines if the Primary Marker is part of a mono-unit system | 1. Mono-unit RGS | RoGI guidelines chap. 3 |

Rouyet et al. ESSD revised manuscript 19/05/2025

| | | 2. Multi-unit RGS | (section 3.2) |
|---|---|---|---|
| | ('Mono-unit RGS') or a multi-unit system ('Multi-unit RGS'). 'Mono-unit RGS': A rock glacier system (RGS) including only one unit. 'Multi-unit RGS': A rock glacier system (RGS) composed of multiple units that are spatially connected, either in a downslope sequence or through coalescence. | | |
| RGS.Primar (O) | Primary ID of the associated Rock Glacier System. RGS + 12 to 15 digits depending on the "Lat.", "Long" values. Always 4 digits after the degrees. | Automatic filling | |
| Morpho. (O) | Defines if the rock glacier identified by the primary marker is a rock glacier with simple or complex morphology. 'Simple': unambiguous and homogeneous morphological expression and/or landcover, connection to the upslope unit and activity (or kinematic if available). 'Complex': ambiguous and heterogenous morphological expression and/or landcover, connection to the upslope unit and activity (or kinematic if available). Despite the spatial variability, there is no sufficient evidence to unambiguously separate units. | 1. Simple 2. Complex | RoGI guidelines chap. 3 (section 3.2) RoGI guidelines chap. 5 (section 5.3) |
| Complet. (O) | Defines if the rock glacier identified by the Primary Marker is completely visible or not. 'No, Ups.Con' means that it is not complete due unclear upslope connection (e.g., several rock glaciers generations are overlapping). 'No, truncated front' means that it is not complete due to truncated front. 'Uncertain' when data or analysis do not allow to decide. | 1. Yes 2. No, unclear connection to the upslope 3. No, truncated front 4. Uncertain | RoGI guidelines chap. 5 (section 5.3) |
| Upsl.Con. (O) | Defines the geomorphological unit directly located upslope of a rock glacier unit or system (five main categories). See related documentation for further information. When dealing with uncertain or intermediate situations, four additional categories are included: 'Poly-connected', 'Other', 'Uncertain' and 'Unknown'. 'Poly-connected': two or more upslope connections (e.g., talus | 1. Talus-connected 2. Glacier forefield-connected 3. Glacier-connected 4. Debris-mantled | RoGI guidelines chap. 3 (section 3.3) RoGI guidelines chap. 5 (section 5.3) |

Rouyet et al. ESSD revised manuscript 19/05/2025

| | | slope-connected |
| --- | --- | --- |
| | and glacier). The use of poly-connected should be restricted to cases where there is no obvious dominance of one connection type.<br><br>'Other': other types of geomorphological sequencing related to a rock glacier landform<br><br>'Uncertain': the geomorphological assessment cannot be performed with confidence.<br><br>'Unknown': the rock glacier unit has been overridden by another one and the former connection to the upslope unit cannot be assessed with confidence anymore. | 5. Landslide-connected<br><br>6. Poly-connected<br><br>7. Other<br><br>8. Uncertain<br><br>9. Unknown | |
| Upsl.Cur. (O) | Defines if the rock glacier is currently connected to the upslope unit or not.<br><br>This attribute is noted only for 'talus-connected' rock glaciers and allows rock glaciers that are currently connected to their upslope unit (i.e., efficient sediment connectivity) to be distinguished from those that have been disconnected from their original source. | 1. Yes<br><br>2. No<br><br>3. Uncertain<br><br>4. Unknown | RoGI guidelines chap. 3 (section 3.3)<br><br>RoGI guidelines chap. 5 (section 5.3) |
| Comment (O) | Comment on possible poly-connection and uncertainty in the geomorphological interpretation. | Text | |
| Acti.Ass. (O) | Defines how the activity assessment was performed: based on geomorphologic evidence only, or with additional kinematic data. | 1. Kinematic<br><br>2. Geomorphologic | RoGI guidelines chap. 3 (section 3.4)<br><br>RoGI guidelines chap. 5 (section 5.3) |
| Acti.Cl. (O) | Activity class assigned to the rock glacier, defined as the efficiency of sediment conveyance (expressed by the surface movement) at the time of observation. See related documentation for further information.<br><br>Already pre-filled if "Kin.Att." is filled.<br><br>It is also possible to change the value manually from the drop-down list, in case of low reliability of the kinematic attribute, e.g., due to unclear pattern in InSAR, unideal slope orientation (N/S) compared to InSAR LOS measurements, or small MA not covering the entire landform. In such cases, the "Kin.Att." may still be documented but assessed as not representative of | 1. Active<br><br>2. Active uncertain<br><br>3. Transitional<br><br>4. Relict uncertain<br><br>5. Relict<br><br>6. Uncertain | RoGI guidelines chap. 3 (section 3.4)<br><br>RoGI guidelines chap. 5 (section 5.3)<br><br>RoGI guidelines chap. 6 (sections 6.1, 6.2 and 6.3) |

| | | | |
|---|---|---|---|
| | the real activity of the rock glacier (based on geomorphologic evidence).<br><br>'Active': rock glacier moving downslope over most of its surface.<br><br>'Active uncertain': the rock glacier unit is not in a relict state, but there is not sufficient data or geomorphological evidence to distinguish between an active and transitional state.<br><br>'Transitional': rock glacier with slow movement only detectable by measurements or movement restricted to areas of non-dominant extent.<br><br>'Relict uncertain': the rock glacier unit is not in an active state, but there is not sufficient data or geomorphological evidence to distinguish between a transitional and relict state.<br><br>'Relict': rock glacier with neither geomorphological evidence nor detection of current movement associated with permafrost creep.<br><br>'Uncertain': the data quality is insufficient to determine any activity status. | | |
| Kin.Att. (O) | Kinematic Attribute (KA) assigned to the rock glacier. The kinematic attribute must be representative of the multi-annual movement rate of the rock glacier unit at the time of an inventory.<br><br>Only if "Acti.Ass." is 'Kinematic'.<br><br>The default category is '0. Undefined'. The rock glacier unit remains in this category when: no (reliable) kinematic information is available, the kinematic information is derived from a single point survey which cannot be related to any MA, the rock glacier unit is mainly characterised by an identified MA of undefined or unreliable velocity, or the kinematic information is too heterogeneous.<br><br>See related documentation on the recommendations to document the KA based on a MA layer. | 0. Undefined<br><br>1. < cm/a<br><br>2. cm/a<br><br>3. cm/y to dm/a<br><br>4. dm/a<br><br>5. dm/a to m/a<br><br>6. m/a<br><br>7. > m/a | RoGI guidelines chap. 6<br>(sections 6.1, 6.2 and 6.3) |
| TypeOfData (O) | Type of data used for kinematic assessment. Use "Kin.Comment" if you want to add more details about the type | Optical<br>Radar | RoGI guidelines chap. 6<br>(sections 6.1, 6.2 and 6.3) |

Rouyet et al. ESSD revised manuscript 19/05/2025

| | | | |
|---|---|---|---|
| | of date used (e.g., InSAR or SAR offset tracking for 'Radar'). Only if "Acti.Ass." is 'Kinematic'. 'Other' can be used if there is a combination of methods (add comments in "Kin.Comment"). | Lidar Geodetic Other | |
| Kin.Period (O) | Period of the data used to assign the KA (e.g., 2018–2020). Only if "Acti.Ass." is 'Kinematic'. | yyyy–yyyy | RoGI guidelines chap. 6 (sections 6.1, 6.2 and 6.3) |
| Destabili. (O) | Describes if the rock glacier unit is (ongoing) or has been (completed) destabilised. Destabilisation refers to rock glaciers with obvious signals of abnormally large displacements, often associated with by the opening of large transversal cracks and/or scarps. 'Yes, ongoing': geomorphological evidence and/or kinematic data signal to an ongoing phase of destabilisation. 'Yes, completed': geomorphological evidence and/or kinematic data confirm a completed destabilisation phase. | 0. No 1. Yes, ongoing 2. Yes, completed 3. Uncertain | RoGI guidelines chap. 3 (section 3.5) RoGI guidelines chap. 5 (section 5.3) |
| Kin.Comment (O) | Comment regarding kinematic information, data type and quality, spatial representativeness, etc. It allows to document uncertainties, especially when the reliability is low or medium. | Text | |
| Rel.Kin. (O) | Reliability of the assignment of the KA based on a qualitative assessment of the data quality and spatial heterogeneity. Only if "Acti.Ass." is 'Kinematic' The attribute accounts for the reliability of MAs covering the rock glacier, the spatial representativeness of the kinematic information (fraction of the rock glacier that is covered by MAs), and the heterogeneity of the available kinematic information (numbers of overlapping MAs with potentially different velocity classes). 'Low': KA assessment is affected by several of the abovelisted limitations. 'Medium': KA assessment is affected by one of the abovelisted limitations. 'High': No limitation is significantly impacting the KA assessment. | 0. Low 1. Medium 2. High | RoGI guidelines chap. 6 (sections 6.1, 6.2 and 6.3) |

| Country (O) | Country Code of the RoGI area. | RO: Romania<br><br>CH: Switzerland<br><br>NO: Norway<br>(T: Troms,<br>F: Finnmark,<br>N: Nordenskiöld<br>Land)<br><br>FR: France<br><br>IT: Italy<br><br>GL: Greenland<br><br>KA: Kazakhstan<br><br>US: U.S.A.<br><br>AR: Argentina<br><br>NZ: New Zealand | See Table 1 and Section 3.3 (naming convention). |
|---|---|---|---|

Rouyet et al. ESSD revised manuscript 19/05/2025

 **Appendix B: Attribute Table of the Moving Area (MA) files**

Attribute table of the Moving Area (MA) gpkg files (M: mandatory attribute; O: optional attribute). The table only includes essential definitions necessary to understand the overall meaning of the attributes. The detailed recommendations for delineating MA based on InSAR and assigning values to each attribute are documented in RGIK (2023b).

| Attribute | Description | Values |
|---|---|---|
| Fid (M) | Unique identifier of the polygon. | Automatic filling |
| MA.ID. (M) | MA + 12 to 15 digits depending on the "Lat.", "Long" values. Always 4 digits after the degrees. (e.g., MA34567S123456E means 3,4567° South and 12,3456° East) | Automatic filling |
| WorkingID (O) | Practical identifier chosen by the operator (e.g., MA_TYR001, TYR002, ... for a moving areas inventory in Tyrol). | Text |
| Ref.PrimaryID (O) | PrimaryID of the related Rock Glacier Unit in the PM attribute table. | Text |
| Vel.Class (M) | Velocity class documenting the overall movement rate observed in a MA during a considered time frame and according to a specific observation time window. It refers to a multi-annual surface velocity representative of the rock glacier creep rate. Using InSAR, it refers to the velocity observed in the radar line-of-sight (LOS) using a dataset covering several months and/or years during a specified observation time window ("Time.Obs."). | 0. Undefined  1. < 1 cm/yr (no movement up to some mm/yr)  2. 1–3 cm/yr (some cm/yr)  3. 3–10 cm/yr  4. 10–30 cm/yr (some dm/yr)  5. 30–100 cm/yr  6. > 100 cm/yr (m/yr and higher) |
| Time.Obs. (O) | Sensor type used to perform the characterisation is documented here. Observation time window (period during which the detection and characterisation is computed/measured, i.e., which months/seasons), and temporal frame (total | Text containing: SENSOR(s) OBSERVATION-TIME-WINDOW TEMPORAL-FRAME  e.g., with InSAR data:  S1 Summer Y1–Y2 (velocity observed from Sentinel-1 with an observation time window in summer, each year between year |

| | | |
|---|---|---|
| | duration during which the periodic measurements/computations are repeated and aggregated for defining the moving area, i.e., which year(s)). | Y1 to year Y2). TSX Summer Y1, Y2, ... (velocity observed from TerraSAR-X with an observation time window in summer, at year Y1, year Y2, etc.) CSK Annual Y1–Y2 (velocity observed from Cosmo-SkyMed with an observation time window of one year, each year in between year Y1 to year Y2). ALOS 08–10 Y1–Y2 (velocity observed from ALOS with an observation time window between August and October each year between year Y1 and year Y2) S1 Summer Y1–Y2 and TSX 10 Y3 (velocity observed from Sentinel 1 with an observation time window in summer, each year between year Y1 to year Y2 and TerraSAR-X with an observation time window centred in October of the year Y3) Note: "Summer" period must be described into the metadata, and it should be at least 2–3 months. |
| Rel.MA (O) | Reliability of the detected moving areas. 'Low': signal interpretation (velocity estimation) and outline are uncertain but there is evidence of movement that needs to be considered. 'Medium': signal interpretation (velocity estimation) or outline is uncertain. 'High': obvious signal and best appropriate configuration (e.g., slope orientation well-aligned with the LOS when using InSAR). | 0. Low 1. Medium 2. High |
| Comment (O) | Comments regarding the MA detection and characterisation (e.g., potential limitations affecting the reliability). | Text (250 characters maximum) |
| Country (O) | Country Code of the RoGI area. | RO: Romania CH: Switzerland NO: Norway (T: Troms, F: Finnmark, N: Nordenskiöld Land) FR: France |

Rouyet et al. ESSD revised manuscript 19/05/2025

| | | IT: Italy |
| | | GL: Greenland |
| | | KA: Kazakhstan |
| | | US: U.S.A. |
| | | AR: Argentina |
| | | NZ: New Zealand |
| | | See Table 1 and Section 3.3 (naming convention). |

**Appendix C: Attribute Table of the Geomorphological Outlines (GO) layers**

Attribute table of the Geomorphological Outlines (GO) gpkg files (M: mandatory attribute; O: optional attribute). The table only includes essential definitions necessary to understand the overall meaning of the attributes. The last column refers to the sections of the RGIK guidelines documenting detailed recommendations for outlining rock glaciers and assigning values to each attribute. At the time of the exercise, the table referred to the sections of several dedicated documents (RGIK, 2022a; 1060 2022b). The RGIK RoGI guidelines have since been merged into one reference document (RGIK, 2023a). The following table has therefore been updated accordingly.

| Attribute | Description | Values | RGIK guidelines |
|---|---|---|---|
| Fid (M) | Unique identifier of the polygon. | Automatic filling | |
| PrimaryID (M) | Unique identifier of the rock glacier unit in the PM attribute table. The digitised polygon in this table is necessarily associated to the previously created Primary Marker (point geometry). The "PrimaryID" must, therefore, be the same as the associated Primary Marker. | Automatic filling | RoGI guidelines chap. 5 (section 5.2) |
| WorkingID (O) | Practical identifier chosen by the operator (e.g., TYR001, TYR002, ... for an inventory in Tyrol). | Text | |
| Out.Type (M) | Outline type. 'Extended': the outline embeds the entire rock glacier up to the rooting zone and includes the external parts (front and lateral margin. 'Restricted': the outline embeds the entire rock glacier up to the rooting zone and excludes the external parts (front and lateral | 1. Extended 2. Restricted 3. Other | RoGI guidelines chap. 3 (section 3.6) RoGI guidelines chap. 5 (section 5.4) |

| | margins). 'Other': if other criteria are applied (to be documented in the Comment field). | | |
|---|---|---|---|
| Rel.Fr. (O) | Reliability of the front outline digitalisation. Qualitative assessment depending on the data quality and the geomorphology complexity of the landform. | 2. High 1. Medium 0. Low | RoGI guidelines chap. 5 (sections 5.4.1 and 5.4.4) |
| Rel.LeftLM (O) | Reliability of the left lateral margin (i.e., orographic perspective) outline digitalisation. Qualitative assessment depending on the data quality and the geomorphology complexity of the landform. | 2. High 1. Medium 0. Low | RoGI guidelines chap. 5 (sections 5.4.2 and 5.4.4) |
| Rel.RightLM (O) | Reliability of the right lateral margin (i.e., orographic perspective) outline digitalisation. Qualitative assessment depending on the data quality and the geomorphology complexity of the landform. | 2. High 1. Medium 0. Low | RoGI guidelines chap. 5 (sections 5.4.2 and 5.4.4) |
| Rel.Ups.Con. (O) | Reliability of the upslope connection outline digitalisation. Qualitative assessment depending on the data quality and the geomorphology complexity of the landform. | 2. High 1. Medium 0. Low | RoGI guidelines chap. 5 (sections 5.4.3 and 5.4.4) |
| Rel.Index (O) | Outline reliability index summing the values assigned to the reliability attributes "RelFr", "Rel.LeftLM", "Rel.RightLM" and "Rel.Ups.Con.". | Automatic filling From 0 (Low) to 8 (High) | RoGI guidelines chap. 5 (section 5.4.4) |
| Comment (O) | Comments regarding the outline, including information regarding the data source(s) used for the delineation and the uncertainties impacting the reliability of the resulting polygon. | Text (250 characters maximum) | |
| Country (O) | Country Code of the RoGI area. | RO: Romania CH: Switzerland NO: Norway (T: Troms, F: Finnmark, N: Nordenskiöld Land) FR: France IT: Italy | See Table 1 and Section 3.3 (naming convention). |

Rouyet et al. ESSD revised manuscript 19/05/2025

| | | GL: Greenland | |
| | | KA: Kazakhstan | |
| | | US: U.S.A. | |
| | | AR: Argentina | |
| | | NZ: New Zealand | |

Rouyet et al. ESSD revised manuscript 19/05/2025