# Peer review of "Rock Glacier Inventories (RoGI) in 12 areas worldwide using a multi-operator consensus-based procedure"

_Earth System Science Data, 2024_

## Referee Comment (RC1)

Review of „Rock Glacier Inventories (RoGI) in 12 areas worldwide using a multi-operator consensus-based procedure "

Author(s): Line Rouyet et al.

MS No.: essd-2024-598

This article presents the results of a first standardized global rock glacier mapping approach. The approach is implemented as a consensus-based multi-operator mapping exercise, involving 41 participants / operators. It uses the recently established guidelines for creating rock glacier inventories (RoGI) developed by the IPA Rock Glacier Inventories and Kinematics (RGIK) community and a standardized QGIS tool to map rock glaciers for 12 different areas around the globe. The dataset includes three GeoPackage files per area: Primary Markers (PM) for rock glacier characterization, Moving Areas (MA) for surface movement detection via InSAR, and Geomorphological Outlines (GO) for delineating rock glacier boundaries.

Such a consistent dataset with rock glacier outlines from different mountain ranges around the globe is of high value for multiple use cases in the future as also nicely described in the conclusion of the manuscript. However, there are few parts which need to be improved in order to be of full use for future users. In the following, general issues of the manuscript and data are described first before adding specific comments on the manuscript.

**General comments – manuscript:**

- **Section 3.3 Output data and naming convention**
  The structure of this section is a bit confusing for an unfamiliar reader. Files with naming convention are used in the first part of the section, before the naming convention itself is introduced and explained in the second part. For a more comprehensive way, I'd suggest starting with the naming convention of the files first and then describe the folder structure using the previously introduced names. Otherwise, please consider avoiding the names of the naming convention in the first part.

- **Section 4 RoGI result description**
  The overall results section has a very comprehensive overview on the results. However, the results are difficult to compare as the size of the investigated area differs quite strongly (ranging from 7 to 82 km²). Therefore, at least one figure should take that into account and provide measures such as RGUs/km² (number of units per square kilometer) or rock glacier area / km² (proportion of entire area covered by rock glaciers), even though the latter might be misleading in cases with a lot of uncertain rock glacier units.
  Another important factor, which is not described anywhere in the manuscript yet, is the availability of different data for each area. Could you incorporate information (maybe in Table 1), which data was available for the mapping exercise (orthophotos (possibly with information about resolution), DEM / hillshade (yes - no)). Further, which type of InSAR data was available (sensor, methodology: stacking, interferograms, PSI) to assess the RGU activity? That should be documented for each area and discussed later, how it is affecting the result (see also comment further down).

- **Section 5.1 Use of the terms "uncertainty" and "reliability"**
  - The term uncertain is used at several different attributes of the PM ("*uncertain rock glacier"* for ambiguous areas; activity attribute). However, it remains partially unclear, how uncertain is defined: Is it uncertain because of lacking data? Or because of diverging opinions of operators? Please specify that more clearly. For the activity attribute it is defined in the guidelines as the former but should be mentioned in the manuscript itself under section 5.1.
  - Similarly, it is mentioned in the discussion that there is the option to document reliability. However, the definition of the reliability should be described in more detail at least in the table in the appendix, possibly also within the manuscript in section 5.1. This should be done for the reliability of the outlines (Appendix C, GO dataset) and the kinematic attribute (Appendix A, PM dataset). They are defined in the guidelines but should be shortly described here to avoid the necessity to search for such definitions in the guidelines. A good example for such a description is the reliability of moving areas in Appendix B, MA dataset.

  Such explanations can be very helpful, when the data is used in the future. This is especially the case for machine learning processes, which can take qualitative measures into account.

- **Section 5.2 Consistency across the different sites**
  - I had a detailed look at the generated results of the different investigated sites and had the impression that there are remaining inconsistencies between the different sites, despite all efforts to reduce them to an absolute minimum. Especially the assignment of certain vs. uncertain landforms seems to be sometimes site-specific and maybe driven by the available data or local knowledge? Could you please add a paragraph or two about that in the discussion?
  - Besides, please include, how the availability of different data (especially different InSAR products) affects the detection and outlining of RGUs for each site individually. So far only the influence on MA is discussed, but it can be expected that with a broader range of InSAR products also potentially more RGUs can be detected and, thereafter, outlined.
  - Further, the quality assessment chapter (5.2) is quite difficult to follow and a lot of site-specific issues are mentioned in the text but hard to get an overview. However, such an overview on the quality of the results for each area could be beneficial to select sites to avoid specific uncertainties or specifically address these issues in the future. Therefore, please elaborate in a more comprehensive way (possibly a Table), which sites are affected by certain issues and which are not.

**General comments – dataset:**

- Disko island data:

There seems to be a projection error with some of the outlines, specifically those towards the east. Or is it a shift in the used WMS layer? Please check units 176 – 183, 194 – 197, 220 and 225.

- Not a rock glacier data:

Specifying and mentioning special landforms as *"Not a rock glacier"* does make sense, when looking at your data. However, this information is only useful, if they have an explanation, why

it is considered to not be a rock glacier. Please add such comments to those (for instance in area 5-1 and 14-1), which do not have it yet, or remove them from the dataset.

**Specific comments:**

L51, L54:
There is a typo in one of the citations: correct citation is Kellerer-Pirklbauer et al. 2024

Table 1:
In the table there is listed the number of **certain final RGU**, which is a result and should therefore not show up in the introduction. Such information is also not necessary at this location.

L54:
*"Conversely, as degradation continues, rock glaciers tend to stabilize and transition progressively into relict landforms"*
Please change "stabilize" to decelerate. Otherwise, it could be understood that all rock glaciers first need to destabilize, which is not the case.

L103-104:
*"Each operator received a common folder including a **similar** dataset organized within a QGIS project (see Section 3.1),"*
Please rephrase the sentence. It should be the same dataset for each operator and site, but probably they differ between the different RoGI areas due to different availability of data. So far, this sentence can be misunderstood.

L401:
*"The assigned KA has contributed to classify the RGU activity as **uncertain (2 RGU) relict (8 RGU)**, transitional (13 RGU), active (20 RGU), and active uncertain (6 RGU)."*
This way of counting is misleading. Please rephrase to "The assigned KA has contributed to classify the RGU activity as **relict (8 RGU), relict uncertain (2 RGU)**, [...]"

L411-414:
*"Rock glacier velocity, on average, was found to increase linearly with elevation up to the 2600–2800 m band, beyond which an inflection occurs, and consistent decimetre annual velocities are attained (Bertone et al., 2024). The activity that characterises rock glaciers in this region below and above 2600 m are consistent, respectively, with transitional and active rock glacier types."*
It is a bit unclear; do you compare your results with previous studies here? If not, consider removing these sentences.

L450:
Remove "Limited". Some of your other sites may also have similar little coverage by research. Therefore, I would not specify it here.

L507:
Remove *"(from LiDAR DEM to filter out the vegetation)"*. A digital terrain model should automatically exclude vegetation. If you want to specifically highlight that vegetation could obscure your results, please consider mentioning it more explicitly.

L531:

*"some are very slowly creeping and so fall into the transitional–relict category"*

Rephrase: some are very slowly creeping and so fall into the transitional or relict category

L547:

Remove *"indeed"*

L627 – 632:

This paragraph very difficult to understand without detailed knowledge on the documentation procedure. Please consider rephrasing it.

---

## Author Response (AR1)

**Answers to Referee #1**

In the following, the comments from the reviewer are in shown in black and our answers are in **bold blue**. References from the manuscript are shown in *italic*. The changes applied to the manuscript (revised version) are underlined. The line numbers refer to the original preprint.

This article presents the results of a first standardized global rock glacier mapping approach. The approach is implemented as a consensus-based multi-operator mapping exercise, involving 41 participants / operators. It uses the recently established guidelines for creating rock glacier inventories (RoGI) developed by the IPA Rock Glacier Inventories and Kinematics (RGIK) community and a standardized QGIS tool to map rock glaciers for 12 different areas around the globe. The dataset includes three GeoPackage files per area: Primary Markers (PM) for rock glacier characterization, Moving Areas (MA) for surface movement detection via InSAR, and Geomorphological Outlines (GO) for delineating rock glacier boundaries.

Such a consistent dataset with rock glacier outlines from different mountain ranges around the globe is of high value for multiple use cases in the future as also nicely described in the conclusion of the manuscript. However, there are few parts which need to be improved in order to be of full use for future users. In the following, general issues of the manuscript and data are described first before adding specific comments on the manuscript.

**Thanks a lot for the positive feedback and the valuable comments to improve the manuscript.**

**General comments – manuscript:**

**Section 3.3 Output data and naming convention**

The structure of this section is a bit confusing for an unfamiliar reader. Files with naming convention are used in the first part of the section, before the naming convention itself is introduced and explained in the second part. For a more comprehensive way, I'd suggest starting with the naming convention of the files first and then describe the folder structure using the previously introduced names. Otherwise, please consider avoiding the names of the naming convention in the first part.

**We understand this comment. We have restructured section 3.3 as suggested, i.e. moving the naming convention before the folder structure:**

*The data package is available on Zenodo (Rouyet et al., 2024; https://doi.org/10.5281/zenodo.14501399). It includes a set of gpkg files organised by areas and product types (PM, MA, GO).*

*The naming convention of each gpkg file follows the product specifications defined by the ESA CCI Permafrost project and is meant to provide a generic structure allowing for updates and/or release of future additional products. All file names follow the same structure: ESACCI-<CCI Project>-<Processing Level>_<Data Type>_<Product String>-<Additional Segregator>_<Layer Type>_-fv<File version>.gpkg*

- *...*

*Accordingly, the data package is structured as followed:*
- ***The folder 'ESACCI-PERMAFROST_ROGI_SINGLE-AREA',** including the RoGI products for each area, for applications focusing on one specific region, with subfolders named as follows:*
  - *...*

- ***The file 'ESACCI-PERMAFROST_ROGI_ALL-AREAS_AOI-PM-MA-GO_2024_fv01.0.gpkg'**, including the AOIs and RoGI results (PM, MA and GO), merged for all areas, for applications requiring the combined use of all inventories.*
- ***The file 'README.pdf' file**, describing the data structure and properties.*

**Section 4 RoGI result description**

The overall results section has a very comprehensive overview on the results. However, the results are difficult to compare as the size of the investigated area differs quite strongly (ranging from 7 to 82 km$^2$). Therefore, at least one figure should take that into account and provide measures such as RGUs/km$^2$ (number of units per square kilometer) or rock glacier area / km$^2$ (proportion of entire area covered by rock glaciers), even though the latter might be misleading in cases with a lot of uncertain rock glacier units.

Another important factor, which is not described anywhere in the manuscript yet, is the availability of different data for each area. Could you incorporate information (maybe in Table 1), which data was available for the mapping exercise (orthophotos (possibly with information about resolution), DEM / hillshade (yes - no)). Further, which type of InSAR data was available (sensor, methodology: stacking, interferograms, PSI) to assess the RGU activity? That should be documented for each area and discussed later, how it is affecting the result (see also comment further down).

**Thanks for these suggestions. Good ideas. We agree with both and have added a new table and a new figure in the revised manuscript. Note that the spatial resolution of the orthomosaics from the WMS is difficult to systematically document, because it varies depending on the area and the scale. We added this sentence in 3.1 (l.179-181):**

- ***QGIS project** structuring the available data and in which the operators performed the work. In addition to the AOI, the InSAR data and initial vector files (gpkg templates), each GIS project incorporated links to Web Map Services (WMS) such as the Google Earth, Bing and ESRI orthomosaics (Table 2). The spatial resolution of such images is typically 0.1-1 m but varies within/across the areas and depending on the scale and zoom levels.*

**Table 2 (placed before Figure 3, in section 3.1):**
*Summary of input data in each RoGI area. The names and locations corresponding to the area numbers are shown in Table 1. The crosses (x) highlight the availability of the corresponding dataset. For InSAR data: the yy-yy numbers correspond to the years available for each InSAR dataset (e.g., 15-19: interferograms or averaged velocity maps between 2015 and 2019).*

| Area number | 5-1 | 6-1 | 7-1 | 8-1 | 9-1 | 10-1 | 11-1 | 12-1 | 13-1 | 14-1 | 15-1 | 16-1 |
|---|---|---|---|---|---|---|---|---|---|---|---|---|
| **Satellite Web Map Services (WMS):** Optical imagery and topographical map | | | | | | | | | | | | |
| Google satellite WMS | x | x | x | x | x | x | x | x | x | x | x | x |
| Bing satellite WMS | x | x | x | x | x | x | x | x | x | x | x | x |
| ESRI satellite WMS | x | x | x | x | x | x | x | x | x | x | x | x |
| OpenTopoMap WMS | x | x | x | x | x | x | x | x | x | x | x | x |
| **Additional optical/thematic data:** HR aerial imagery and national topographical map | | | | | | | | | | | | |
| Extra HR aerial image | x | x | x | x | x | x | | | | | | |
| National topo. map | | x | x | x | | x | | | | | | x |
| **DEM products:** Low/High-Resolution (LR/HR) DEM and/or associated products (e.g., hillshades, slope, aspect) | | | | | | | | | | | | |
| LR DEM (10–30m) | x | x | x | x | x | x | x | x | x | x | x | x |
| HR DEM (< 10m) | x | x | | | | x | x | | | | | |
| **InSAR data:** Wrapped interferograms (ifgs) and velocity maps from Stacking and Persistent Scatterer Interferometry (PSI) | | | | | | | | | | | | |
| Sentinel-1 ifgs | 16-19 | 17-19 | 17-19 | 17-20 | 18-20 | 16-19 | 18-19 | 15-19 | 15-19 | 16-19 | 18-20 | 15-23 |
| ERS-1/2 ifgs | | | | | | | | | 98-99 | 91-95 | | |
| ALOS-1 ifgs | | | | | | 07-10 | 07-10 | | 06-10 | 06-09 | 08-11 | 07-08 |

| | | | | | | | | | | | | |
|---|---|---|---|---|---|---|---|---|---|---|---|---|
| ALOS-2 ifgs | 14-19 | 14-21 | | | | | | 15-17 | 14-16 | 15-16 | 16-19 | |
| SAOCOM ifgs | | 21 | | | | | | | | | 21-22 | 21-23 |
| Cosmo-SkyMed ifgs | | | | | | 16-20 | | | | | | |
| TerraSAR-X ifgs | | 09-14 | | | | | | | | | | |
| 6–12d ifgs Stacking | | 19 | 15-19 | 15-20 | 15-20 | 18-19 | 18-19 | 18 | 18 | 18-19 | 18-19 | 18 |
| Combined 6d–annual ifgs Stacking | | | 15-19 | 15-20 | 15-20 | | | | | | | |
| PSI | 15-21 | | | 15-19 | | | | | | | | |

**Figure 9 (placed after Figure 8, in section 4):**

*Density of rock glaciers in the studied areas. The grey bars show the percent of the area covered by rock glaciers, according to the mapped extended outlines. The corresponding values are shown on the primary vertical axis on the left. The black symbols (dots: certain rock glaciers; triangles: certain and uncertain rock glaciers) show the numbers of identified RGU in respect to the size of the area (number per km2). The corresponding values are shown on the secondary vertical axis on the right. The area numbers and the acronyms of the corresponding countries are similar to Figure 5 and according to Table 1 naming convention.*

[Figure]

**The last paragraph of the overall result description (section 4, l.324-327) has been modified as followed:**

*Based on the extended outlines, the RGU have a typical size ranging between 0.01 and 0.25 km² (median value of each area, Figure 8). The boxplots indicate large differences in size between and within the areas. It should be noted that in areas dominated by large rock glaciers (e.g., area 12-1 GL; area 13-1 KA), small talus-connected rock glaciers may have been overlooked. The size of the areas significantly varies (ranging from 7 to 82 km², see Table 1). The size of the mapped landforms, as well as the number of certain and uncertain RGU, in respect to the size of the area, are also highly variable (Figure 9). Some areas are characterised by many small landforms (e.g., area 6-1 CH; area 16-1 NZ), while others are dominated by few large rock glacier units (e.g., area 12-1 GL; area 13-1 KA).*

**Section 5.1 Use of the terms "uncertainty" and "reliability"** The term uncertain is used at several different attributes of the PM ("*uncertain rock glacier*" for ambiguous areas; activity attribute). However, it remains partially unclear, how uncertain is defined: Is it uncertain because of lacking data? Or because of diverging opinions of operators? Please specify that more clearly. For the activity attribute it is defined in the guidelines as the former but should be mentioned in the manuscript itself under section 5.1.

Similarly, it is mentioned in the discussion that there is the option to document reliability. However, the definition of the reliability should be described in more detail at least in the table in the appendix, possibly also within the manuscript in section 5.1. This should be done for the reliability of the outlines (Appendix C, GO dataset) and the kinematic attribute (Appendix A, PM dataset). They are defined in the guidelines but should be shortly described here to avoid the necessity to search for such definitions in the guidelines. A good example for such a description is the reliability of moving areas in Appendix B, MA dataset.

Such explanations can be very helpful, when the data is used in the future. This is especially the case for machine learning processes, which can take qualitative measures into account.

**Thanks for the comment. There are different options to document the uncertainty and reliability, at the different steps of the work. These are all qualitative measures based on criteria that varies depending on the attribute. These possibilities were summarised in section 5.1 (l.483-498) and briefly listed in the attribute tables in the appendixes. We initially decided to avoid incorporating too many elements from the reference documents in the paper and referred to the corresponding sections of the RGIK RoGI guidelines, but the comments from both reviewers showed the need to add more information about these definitions and criteria. We therefore made several modifications to clarify these points in several parts of the manuscript. Some examples in the following:**

**In section 2.2, we added at l.128:** *When combining the results between operators, the team agreed on which units were categorised as "certain" or "uncertain" within each area. In some cases, the rock glaciers remained "uncertain" when there was not enough evidence that the landform is a rock glacier, or when the team decided that the landform was too complex to be accurately characterised and outlined with the currently available data. Keeping an information about the location of these uncertain landforms may allow for future updates if new data is becoming available.*

**In section 5.1, the paragraph is modified as followed:** *In the attribute tables of the three GeoPackage files, various fields document the reliability of the mapping and morpho-kinematic assessment, according to identified uncertainties and limitations:*
- *For the PM files, an attribute "uncertain" describes ambiguous areas that should be investigated in the future (need for additional data and/or field visit). For educational purposes, an attribute "not a rock glacier" could also be used to highlight landforms that are likely to be misinterpreted as rock glaciers. The level of uncertainty and complexity can be highlighted for many morpho-kinematic attributes, either in the selectable categories (for example "active uncertain", "transitional uncertain", and "relict uncertain" for the attribute "Activity") or using an additional reliability attribute for the kinematic assessment (Appendix A). Additional comments describing the uncertainty sources and ambiguities in the interpretation can be written in two "Comment" fields.*
- *For the MA files, the reliability (or the degree of confidence) of the results is qualitatively documented in accordance with the quality of the detection, the delineation of the*

> *Moving Areas based on the available InSAR data, the signal interpretation and the* *resulting velocity estimation (Appendix B)**. When medium–low reliability is set (uncertain InSAR signal and/or unclear MA outlines), information on the uncertainty sources* *and ambiguities in the interpretation* *can be described in a "Comment" field.*

- *For the GO files, the reliability of the delineation at different locations of the rock glacier (front, left/right lateral margins, upslope boundary) is estimated with a score of 0 (low), 1 (medium), or 2 (high).* *It consists of a qualitative assessment depending on the data quality and the geomorphology complexity of the landform (Appendix C).* *The* *automatic summation of the scores (0–8) gives a general estimate of the outline reliability for the entire landform.* *Information regarding the data source(s) used for the delineation and the uncertainties impacting the reliability of the resulting polygon can be documented in a "Comment" field.*

**We agree that there were discrepancies in the level of details to explain the attributes in the tables in the appendixes. In the revised version, we added more detailed description of several fields. For the PM product, we included the definition of mono/multi-unit RGS, simple and complex morphology, the definition of the activity classes, incl. the meaning of 'active uncertain' and 'relict uncertain', the definition of 'poly-connected', 'other', 'uncertain' and 'unknown' for the upslope connection, the definition of the kinematic attribute, incl. explanations on when it should remain 'undefined' and explanations regarding the different levels of reliability. For the MA product, we added a more detailed explanation about the velocity classes. For the GO product, we included the definitions of the outline types and explanations on the criteria to document the reliability. Due to the many changes, we are not incorporating all tables in this answer, but these will be included in the revised submission.**

**Section 5.2 Consistency across the different sites** I had a detailed look at the generated results of the different investigated sites and had the impression that there are remaining inconsistencies between the different sites, despite all efforts to reduce them to an absolute minimum. Especially the assignment of certain vs. uncertain landforms seems to be sometimes site-specific and maybe driven by the available data or local knowledge? Could you please add a paragraph or two about that in the discussion?

Besides, please include, how the availability of different data (especially different InSAR products) affects the detection and outlining of RGUs for each site individually. So far only the influence on MA is discussed, but it can be expected that with a broader range of InSAR products also potentially more RGUs can be detected and, thereafter, outlined.

Further, the quality assessment chapter (5.2) is quite difficult to follow and a lot of site-specific issues are mentioned in the text but hard to get an overview. However, such an overview on the quality of the results for each area could be beneficial to select sites to avoid specific uncertainties or specifically address these issues in the future. Therefore, please elaborate in a more comprehensive way (possibly a Table), which sites are affected by certain issues and which are not.

**It is correct to say that inconsistencies still remain between the different regions, due to different levels of geomorphological complexity, variable data quality and local knowledge. In order to better acknowledge this point, we modified the first paragraph of section 5.2:**
*Here we summarize the observations about the* *uncertainties and limitations* *of the three output files, based on the results in the 12 areas and the feedback of the operator teams.* *Most*

*challenges are common for all areas, while a few are affecting specific areas only. The main identified uncertainties and limitations of each output product are described in the following sections and summarised in Table 3. Despite the effort to standardise the procedure and reduce the differences between the areas, we acknowledge that discrepancies remain in the final products. These are due to the different levels of geomorphological complexity, the variable numbers of landforms and the density of their distribution, as well as the heterogenous data quality, local knowledge and research history. In case of operator discrepancies, the decisions were taken at the team level, ensuring homogeneity within each area. Major questions were discussed during PI coordination meetings and communicated across the teams thanks to operators working in several areas. The parallel timeline of the work in all areas contributed to a good communication on the common challenges, but did not discard all risks of inter-regional differences and subjective treatment.*

**We added one bullet point in 5.2.1:**
*InSAR was useful for detecting rock glaciers that may have been missed when only looking at optical images. However, it also added an additional source of variability between the regions, because the data availability and properties vary from a region to another. In areas with multi-temporal PS/DS InSAR data, the detection capability to low velocity was increased. In areas with X-band SAR data, interferograms with higher spatial resolution were provided, which allowed for detecting smaller moving landforms. In areas with L-band SAR data and 6-days Sentinel-1 repeat-pass, the maximal detection capability was increased.*

**We modified the last bullet point in 5.2.2:**
*A general challenge with InSAR analysis is to ensure that the detected movement is representative of the rock glacier creep rate and not significantly affected by other processes (e.g., landslide, solifluction, thaw subsidence). Analysing a diverse set of interferograms (various SAR geometries, time intervals, months and years) allow to reduce the risk of misinterpretation by providing complementary information about the spatial and temporal characteristics of the movement pattern. However, we cannot fully discard the possibility that some MA identified on a rock glacier might be affected by other processes. In case of rapid permafrost degradation, the detected movement may correspond to a mixed signal from downslope creep and subsidence due to ice core melt. In cold regions with continuous permafrost, the ground is highly dynamic during the thawing season, which makes it difficult to dissociate the InSAR signal on the rock glacier from surroundings areas that also move. When analysing small and slowly creeping talus-connected rock glaciers, it was sometimes challenging to discriminate the movement associated with rock glacier creep from other processes, such as thaw subsidence in ice-rich lowlands located directly at the foot of the mountain ridges.*

**In the initial version, we added the references to specific regions in parentheses (e.g., area xx-x) in the sections 5.2.1-5.2.3. These references were meant to exemplify where the limitations had been reported by the RoGI team (i.e., not an exhaustive list). As suggested, we have tried to summarise the limitations in a table, placed at the end of section 5.2. We have removed the examples in parentheses to provide a lighter version of the text.**

**Table 3.** *Overview of the main uncertainties and limitations of the RoGI products and how they apply to the 12 areas. The crosses (X) show where the problem has been explicitly reported by the RoGI team/PI. The circles (O) show where the problem might happen for specific landforms, but had not been reported has a main limitation by the RoGI team/PI. The area numbers are similar to Figure 5 and according to Table 1 naming convention.*

| | 5-1 | 6-1 | 7-1 | 8-1 | 9-1 | 10-1 | 11-1 | 12-1 | 13-1 | 14-1 | 15-1 | 16-1 |
|---|---|---|---|---|---|---|---|---|---|---|---|---|
| **PM detection and characterisation** | | | | | | | | | | | | |
| Optical imagery affected by shadows, clouds, snow | O | O | O | O | X | O | O | X | O | X | O | O |
| Dense vegetation cover on relict rock glaciers | X | | | | | | | | | | | |
| Dominance of large RGU and small RGU likely overlooked | | | | | | | | O | X | X | X | |
| Ambiguous imbrication of periglacial landforms | O | O | X | X | O | X | O | O | O | X | O | |
| Ambiguous rock glacier and glacier/forefield continuum | O | X | O | O | O | O | O | O | O | X | X | X |
| Variable categorisation of landslide-connected RGU | O | O | X | O | O | X | O | O | O | O | O | |
| Difficulty to select of RGU for complex multi-unit systems | O | O | X | O | O | O | O | O | X | X | X | O |
| Ambiguity in activity in Arctic cold regions with slow/no MA | | | | | X | | O | | O | | | |
| Difficulty to discriminate active/transitional | | O | O | | O | O | O | O | O | O | O | X |
| **MA detection, delineation and characterisation** | | | | | | | | | | | | |
| Fewer available Sentinel-1 images and longer repeat-pass | | | | | | | | X | X | X | X | X |
| Challenge of velocity estimate for operators with little InSAR experience | X | X | X | X | X | X | X | X | X | X | X | X |
| Difficulty to document and interpret small and slow MA | X | O | O | X | O | O | O | O | O | O | O | O |
| Difficulty to discriminate creep from other processes | O | O | O | X | X | O | O | X | O | X | O | O |
| **GO delineation and characterisation** | | | | | | | | | | | | |
| Uncertainty in the delineation of the upper boundaries | X | X | X | X | X | X | X | X | X | X | X | O |
| Uncertainty in the delineation of eroded, reworked or exaggerated fronts | X | X | O | X | O | X | X | O | O | X | X | O |
| Unclear lateral margins for small rock glaciers | O | O | O | X | O | O | X | O | O | O | O | O |
| Difficulties to outline complex RGS with multiple RGU | O | O | X | O | O | O | O | O | X | X | X | O |
| Variable quality of optical imagery and georeferencing shifts | O | O | O | X | X | O | O | X | X | X | O | O |

**General comments – dataset:**

- Disko island data:
  There seems to be a projection error with some of the outlines, specifically those towards the east. Or is it a shift in the used WMS layer? Please check units 176 – 183, 194 – 197, 220 and 225.

  **Thanks for having seen it. This shift is indeed due to low quality and georeferencing shifts in the online data at the time of the delineation. In that case, it does not impact the whole area similarly. We have therefore corrected the dataset to show consistency. We have mentioned this recent change in the 'Comment' field of the GO layer. The new version will be submitted on Zenodo at the same time as the revised manuscript.**

  **Note that some shifts may have occurred in other regions. This issue is explained in the last bullet point of section 5.2.3. We completed the paragraph by adding the following sentences:** *The data source used for the final outlines and the time it applies is specified in the attribute table. The results apply for the period during which the outlines were drawn. If viewing the results with a more recent background imagery, new shifts may occur due to the regular updates of the WMS data sources and their variable quality.* **We have corrected the GO data files to always include a time reference in the 'Comment' field (in addition to the data source).**

- Not a rock glacier data:
  Specifying and mentioning special landforms as *"Not a rock glacier"* does make sense, when looking at your data. However, this information is only useful, if they have an explanation, why it is considered to not be a rock glacier. Please add such comments to those (for instance in area 5-1 and 14-1), which do not have it yet, or remove them from the dataset.

  **Thanks for the comment. We agree. These were oversights. We have modified the dataset to remove some (when we thought they were unnecessary) or include an explanation (when we thought they were useful). A new version will be submitted on Zenodo at the same time as the revised manuscript.**

**Specific comments:**

L51, L54:
There is a typo in one of the citations: correct citation is Kellerer-Pirklbauer et al. 2024

**Thanks. Corrected.**

Table 1:
In the table there is listed the number of **certain final RGU**, which is a result and should therefore not show up in the introduction. Such information is also not necessary at this location.

**Indeed. We removed it and replaced by information about the elevation range in the AOI.**

L54:
*"Conversely, as degradation continues, rock glaciers tend to stabilize and transition progressively into relict landforms"*

Please change "stabilize" to decelerate. Otherwise, it could be understood that all rock glaciers first need to destabilize, which is not the case.

**Agree. Modified as suggested.**

L103-104:
*"Each operator received a common folder including a **similar** dataset organized within a QGIS project (see Section 3.1),"*

Please rephrase the sentence. It should be the same dataset for each operator and site, but probably they differ between the different RoGI areas due to different availability of data. So far, this sentence can be misunderstood.

**Yes, as written at l.102, we describe here the data structure for each area (i.e. within each team). We clarified this point by rephrasing the sentence at l.103-104:**

*Within each team, each operator received a common folder including a similar dataset applicable for the area. The input data is organised within a QGIS project…*

***We made two additional modifications in Section 2.1 to make extra clear that the 12 teams correspond to the 12 areas (each team corresponds to one area):***

*A Principal Investigator (PI) was designated to coordinate the work of the inventory team in each area.* (l.89-90)

*…coordinating the work between the 12 teams, corresponding to the 12 areas.* (l.100)

L401:
*"The assigned KA has contributed to classify the RGU activity as **uncertain (2 RGU) relict (8 RGU)**, transitional (13 RGU), active (20 RGU), and active uncertain (6 RGU)."*

This way of counting is misleading. Please rephrase to "The assigned KA has contributed to classify the RGU activity as **relict (8 RGU), relict uncertain (2 RGU)**, […]"

**Agree. Modified.**

L411-414:
*"Rock glacier velocity, on average, was found to increase linearly with elevation up to the 2600–2800 m band, beyond which an inflection occurs, and consistent decimetre annual velocities are attained (Bertone et al., 2024). The activity that characterises rock glaciers in this region below and above 2600 m are consistent, respectively, with transitional and active rock glacier types."*

It is a bit unclear; do you compare your results with previous studies here? If not, consider removing these sentences.

**All 4.x subsections are organised similarly for each area: a first paragraph summarising the general geographic setting; a second paragraph highlighting mountain permafrost background in the area and if available, past research on rock glaciers; and a third paragraph focusing on the actual results from the multi-operator RoGI exercise.**

**For Southern Venosta, l.411-414 are describing results from past research, while l.418-421 summarise the results of the current study. The differences between both paragraphs may have been unclear cause both studies are quite similar, and the results are overall consistent. We therefore simplified the 2nd paragraph (l.418-414) and rephrased to clarify:**

*According to a recently compiled geomorphological inventory, the area is characterised by the highest rock glacier density within South Tyrol (~ 1.1 #/km$^2$ against a regional average of 0.54 #/km$^2$) (Scotti et al., 2024). Subsequent integration of this geomorphological inventory with InSAR-based kinematic information across the Southern Venosta subregion led to detect 375 intact and 428 relict rock glaciers (Bertone et al., 2024). On average, the velocity of intact rock glaciers was found to increase linearly with elevation up to the 2600–2800 m band (where MAAT declines from about -1 to -2 °C), beyond which a kinematic plateau occurs. This band marks a broad altitudinal shift from transitional (< dm/yr) to active (> dm/yr) rock glacier types (Bertone et al., 2024).*

L450:
Remove "Limited". Some of your other sites may also have similar little coverage by research. Therefore, I would not specify it here.

**Agree. Modified.**

L507:
Remove *"(from LiDAR DEM to filter out the vegetation)"*. A digital terrain model should automatically exclude vegetation. If you want to specifically highlight that vegetation could obscure your results, please consider mentioning it more explicitly.

**It was indeed the main point of this sentence (when considering the main data source for the mapping: optical imagery from passive sensors). But in addition, we meant to say that in such cases, the use of complementary input data (HR DEM) helps overcoming the problem, which is especially valuable in areas dominated by relict landforms with dense vegetation cover.**

**We rephrased l. 505-507 as followed:**
*Warm regions in the marginal permafrost zone can be dominated by relict landforms with dense vegetation cover that may hinder detailed mapping based on passive optical remote sensing only. In such areas, terrain hillshades from high-resolution LiDAR DEM are highly valuable as complementary input data. The availability and quality of such products are however variable from a region to another.*

L531:
*"some are very slowly creeping and so fall into the transitional–relict category"*
Rephrase: some are very slowly creeping and so fall into the transitional or relict category

**Corrected.**

L547:
Remove *"indeed"*

**Removed.**

L627 – 632:
This paragraph very difficult to understand without detailed knowledge on the documentation procedure. Please consider rephrasing it.

**We tried to clarify this paragraph as followed:**
*The assignment of the activity attribute based on geomorphological and/or kinematic criteria requires clarification in the guidelines. The InSAR analysis led to the generation of a MA layer with polygons highlighting where movement has been detected. For characterising the kinematics*

*and the activity, the operators used the MA layer as input. However, some rock glaciers are not covered by any MA. In such cases, it was recommended to avoid overinterpreting the absence of detected movement, because a rock glacier without any MA may mean two different things: 1) there is no movement or too low to be detected, 2) the data quality and/or coverage did not allow for detecting it. In such cases, some operators only focused on geomorphological criteria to assign the activity. A kinematic attribute with low velocity and low reliability index may have been documented, but was not used to set the activity. For other operators, the lack of movement evidence has been used in synergy with geomorphological evidence, as an additional indicator confirming the geomorphological interpretation.*

**Answers to Referee #2**

In the following, the comments from the reviewer are in shown in black and our answers are in **bold blue**. References from the manuscript are shown in *italic*. The changes applied to the manuscript (revised version) are underlined. The line numbers refer to the original preprint.

I enjoyed reading this paper, and to my understanding this approach of mapping rock glaciers based on a given set of assumptions and definitions seem to be well founded and carried out in a thought through manner. I find the considerations accounted for in the manuscript to be meaningful and interesting, and the paper seems like a natural next step following the work carried out by the RGIK group. There are a few things I think could improve the manuscript a bit, and this would mainly be for readability and overview.

**Thanks a lot for the positive feedback and the valuable comments to improve the manuscript.**

**Section 2.1:** you describe a bit how the exercise was performed, but could you perhaps elaborate a bit here? For instance, were mapping teams assembled randomly, or did they comprise individuals with local expertise? Table 1 suggests that the principal investigator of each team possesses some or significant area-specific knowledge, but what about other team members? From my own background I know that it can be quite difficult to 1. change my opinion about familiar terrain, and 2. communicate my interpretation of landforms to local experts in areas that I lack familiarity with. I assume that you found ways to handle this, but it would be nice to know.

**Thanks for the comment. A sentence mentioned (partly) this point in section 2.2 (l.100-102), but this was somewhat misplaced. We moved it to section 2.1 and added new information about the way the teams have been assembled:**

*A Principal Investigator (PI) was designated to coordinate the work of the inventory team in each area. All PIs had past or ongoing research in the area they were leading. The volunteer operators were found within the involved institutions and after a call for participation in June 2023 using the RGIK mailing list (about 200 subscribers). The participants were free to choose in one or more area(s) to perform the work, depending on their interest and time availability. To ensure enough operators in each area, as well as a diversity of geographical background, competence and seniority, members of the PI team acted as operators in areas where few people signed up. The resulting inventory teams were composed of five to ten operators (including the PI; Table 1). Some operators worked in several areas. One operator (R. Delaloye) performed the work in all the areas, which helped communicating common challenges and coordinating key decisions across the teams. The exercise involved a total of 41 persons (see Author list and Acknowledgments).*

In **table 1** you include the numbers of RGUs within each area, this is really a result. Instead, consider including information on the materials available in each study area, such as orthophotos (with resolution), DEMs (with resolution), and InSAR quality and availability. This extra information might fit better in chapter 3 after you go through the content of the input data.

**Thanks for the comment, which overlaps with a similar suggestion from reviewer 1. We have added a new table 2 for input data (see below). We removed the number of certain RGU in table 1, and replaced it by information about the elevation range in the AOI.**

**Table 2 (placed before Figure 3, in section 3.1):**

*Summary of input data in each RoGI area. The names and locations corresponding to the area numbers are shown in Table 1. The crosses (x) highlight the availability of the corresponding dataset. For InSAR data: the yy-yy numbers correspond to the years available for each InSAR dataset (e.g., 15-19: interferograms or averaged velocity maps between 2015 and 2019).*

| Area number | 5-1 | 6-1 | 7-1 | 8-1 | 9-1 | 10-1 | 11-1 | 12-1 | 13-1 | 14-1 | 15-1 | 16-1 |
|---|---|---|---|---|---|---|---|---|---|---|---|---|
| **Satellite Web Map Services (WMS):** Optical imagery and topographical map | | | | | | | | | | | | |
| Google satellite WMS | x | x | x | x | x | x | x | x | x | x | x | x |
| Bing satellite WMS | x | x | x | x | x | x | x | x | x | x | x | x |
| ESRI satellite WMS | x | x | x | x | x | x | x | x | x | x | x | x |
| OpenTopoMap WMS | x | x | x | x | x | x | x | x | x | x | x | x |
| **Additional optical/thematic data:** HR aerial imagery and national topographical map | | | | | | | | | | | | |
| Extra HR aerial image | x | x | x | x | x | x | | | | | | |
| National topo. map | | x | x | x | | x | | | | | | x |
| **DEM products:** Low/High-Resolution (LR/HR) DEM and/or associated products (e.g., hillshades, slope, aspect) | | | | | | | | | | | | |
| LR DEM (10–30m) | x | x | x | x | x | x | x | x | x | x | x | x |
| HR DEM (< 10m) | x | x | | | | x | x | | | | | |
| **InSAR data:** Wrapped interferograms (ifgs) and velocity maps from Stacking and Persistent Scatterer Interferometry (PSI) | | | | | | | | | | | | |
| Sentinel-1 ifgs | 16-19 | 17-19 | 17-19 | 17-20 | 18-20 | 16-19 | 18-19 | 15-19 | 15-19 | 16-19 | 18-20 | 15-23 |
| ERS-1/2 ifgs | | | | | | | | | 98-99 | 91-95 | | |
| ALOS-1 ifgs | | | | | | 07-10 | 07-10 | | 06-10 | 06-09 | 08-11 | 07-08 |
| ALOS-2 ifgs | 14-19 | 14-21 | | | | | | 15-17 | 14-16 | 15-16 | 16-19 | |
| SAOCOM ifgs | | 21 | | | | | | | | | 21-22 | 21-23 |
| Cosmo-SkyMed ifgs | | | | | | | 16-20 | | | | | |
| TerraSAR-X ifgs | | 09-14 | | | | | | | | | | |
| 6–12d ifgs Stacking | | 19 | 15-19 | 15-20 | 15-20 | 18-19 | 18-19 | 18 | 18 | 18-19 | 18-19 | 18 |
| Combined 6d–annual ifgs Stacking | | | 15-19 | 15-20 | 15-20 | | | | | | | |
| PSI | 15-21 | | | 15-19 | | | | | | | | |

**Figure 3**: The text and symbols are quite small, making it difficult to read.

**Thanks for the comment. We made a new version, which is hopefully easier to read:**

[Figure]

***Figure 3.** Example of QGIS data structure and dialog box for semi-automatic attribute filling in area 13-1 (Northern Tien Shan, Kazakhstan). An example of Sentinel-1 wrapped interferogram is displayed within the AOI extent. The boundaries of the RoGI area (black polygon), the PM (white dots and triangles), and the MA (yellow to red polygons) are displayed as top layers. For sake of*

*visualisation, the GO layer is not shown. See example with GO in Figure 4. Background map: ESRI Satellite Web Map Service.*

**Figure 4**: The legend contains an entry for "not a rock glacier," which is not previously mentioned in the manuscript. It is acknowledged in Section 5.1 as potentially misinterpreted landforms, serving as an educational element. However, it may be more appropriate to exclude this annotation from the figures. Instead, incorporating it into attribute tables or supplementary materials like you have done could facilitate learning, so it might be useful to keep.

**Agree. Removed in this figure. The details/explanations about why a landform is categorised as "not a rock glacier" are included in the PM attribute tables. In addition, we noticed that the current figure did not include the PM of all operators (black dots/triangles), which was an error. This has been corrected.**

[Figure]

**Figure 4.** *Example of RoGI results in part of area 7-1 NO-T (Troms, Norway), showing individual operator results and final consensus-based results (Primary Markers: PM; Geomorphological Outlines: GO). For sake of visualisation, the MA layer is not shown, but was used to assign the PM kinematic attribute displayed here with a green–red colour scale. See example with MA in Figure 3. Background: NorgeiBilde orthophoto (2016-08-2016).*

**Chapter 4**: Consider restructuring this chapter to begin with the description of study areas (Section 4.1 onward) before presenting Figures 5-8. This change could enhance the comprehension and interpretation of the figures.

**Agree. We changed the structure as suggested, adding a section 4.13 (Results summary across all areas), after the description of each area (sections 4.1-4.12). We added the following sentence in the introduction of Section 4 (l.287): "***In the following, we describe the results in each area separately (Sections 4.1–4.12) before summarising the findings across all areas (Section 4.13).***"**

I appreciate how challenges and uncertainties across regions and operators are summarized, reflecting the inherent difficulties in interpreting geomorphology in certain areas. I also like that you address some issues connected to how we traditionally have interpreted intact landforms as active, while in the kinematical and more recent definitions these are considered transitional or even relict. Maybe such observations could challenge the value we add to the current movement rates in high-arctic areas.

In general, I especially enjoyed reading the quality assessments of the different products mapped, i.e. the results of this exercise. I think you go through the different points carefully and thoroughly, and I gained some new insights while reading.

**Thanks a lot for this positive feedback, much appreciated!**

**Map material:**

To me this looks good, and with some help from the descriptions in the appendices it was quite easy to navigate in the mapped material.

From the text, I cannot read whether you did some "user sensitivity" tests or what to call it. It appears there are notable differences between the extended and restricted rock glacier outlines between some regions. Were systematic, regional differences in extended and restricted RG areas assessed? You mention this a bit in sec. 5.2.1. and 5.2.3., and maybe this issue is mainly addressed in the attribute tables as low outline reliability. However, when I had a look at one of the rock glacier outlines (RGU707506N277873E) in Finnmark where there is a rather large difference between the restricted and the extended outline of the front, the extended front position is marked with 2 (high reliability) while the restricted front position is marked with 1 (medium reliability), while both have 0 (low reliability) connected to their upslope margins. The uncertainties of the upslope margins are well accounted for in the text, but from the mapped material to me it looks like it is the front positions that are uncertain in this specific case. I had a look at the other rock glaciers in the same area, and the ones I had a look at seem to be classified in the same way. (I only looked into a few in the vicinity of RGU707506N277873E.)

**We have not performed a systematic inter-regional sensitivity analysis of the GO outlining process. Despite the despite the effort to standardise the procedure and reduce the differences between the regions, we acknowledge that some discrepancies remain. These are due to the different (and partly subjective) choices of the teams but also to real geomorphological differences between the areas. In accordance with a similar comment from reviewer 1, we modified the start of section 5.2:**

*Here we summarise the observations about the uncertainties and limitations of the three output files, based on the results in the 12 areas and the feedback of the operator teams. Most challenges are common for all areas, while a few are affecting specific areas only. The main identified uncertainties and limitations of each output product are described in the following sections and summarised in Table 3. Despite the effort to standardise the procedure and reduce the differences between the areas, we acknowledge that discrepancies remain in the final products. These are due to the different levels of geomorphological complexity, the variable numbers of landforms and the density of their distribution, as well as the heterogenous data quality, local knowledge and research history. In case of operator discrepancies, the decisions were taken at the team level, ensuring homogeneity within each area. Major questions were discussed PI coordination meetings and communicated across the teams thanks to operators working in several areas. The parallel timeline of the work in all areas contributed to a good*

*communication on the common challenges, but did not discard all risks of inter-regional differences and subjective treatment.*

**We had partly mentioned the question of the uncertainty of the front delineation in the second bullet point of 5.2.3, but agree it was less discussed compared to the upper boundary. The case you described in Finmark is a landform with a quite smoothed front, for which the location of the restricted outline is ambiguous, due to rounded topography. We have made the following modification in 5.2.3 to include this point:**

*In some cases, the delineation of the front was challenging, especially if the toe of the rock glacier was reworked by other processes, such as solifluction. Smooth fronts and rounded ridges and furrows, often associated with relict and transitional landforms, may lead to ambiguous delineation of the restricted outlines. For a rock glacier developing on a steep slope, the front may also be difficult to distinguish. Some problems were for instance identified in cases of exaggerated fronts blended with the downside talus slope. Small rock glaciers, such as debris-mantled-connected rock glaciers, or embryonic talus-connected landforms (protalus ramparts) often had ambiguous lateral margins, challenging for outlining. Such complicated cases were discussed during team meetings to find a mutually agreeable solution. When the location of the boundary was uncertain, the front and/or lateral outline reliability was set to "low" or "medium" in the attribute table.*

Additionally, I am a bit confused by the discrepancies between rock glacier outlines and MA polygons. Could you clarify why these sometimes overlap and other times do not? While only "certain" rock glaciers are outlined, many polygons with MA values are neither marked as "uncertain" nor as "not a rock glacier" in the mapped material. Conversely, there are instances where RG outlines exist without corresponding MA polygons (e.g. Disko, Greenland).

**The moving area detection/delineation is based InSAR only. The polygons are not necessarily following the landform margins. It only shows where movement has been detected. If the movement is heterogenous and/or if InSAR is affected by limitations (data gap, underestimation due to slope orientation diverging from the radar viewing angle, decorrelation due to snow, etc.), the MA polygon may only be partly overlapping with the rock glacier.**

**The MA step is performed in parallel to the PM step. Partly iteratively (cause InSAR may help to detect new landforms) but partly independently considering that the MA outlines were delineated before the team decision on the final PM selection. It means that when it was a doubt at this stage (rock glacier, not a rock glacier, uncertain?), the operators may have decided to draw a MA. If the associated landform was discarded after the team discussion (not a rock glacier or uncertain), the MA remained (information of movement but not associated to a certain rock glacier). We believed it would have been a waste to remove the information, even if some MA are not related to a rock glacier but something else, e.g. solifluction or landslide. That could be of interest as well! In Finnmark, for instance, we comprehensively mapped the movement within the entire area considering the complex interaction of several periglacial processes. We kept this information although several MAs were not further used in the next steps of the exercise.**

**Your question showed the need to add more explanations on the way InSAR has been used. We therefore extended the description of the MA step, in section 2.2:**

***Detect, delineate, and classify Moving Areas (MA) using InSAR.*** *This task was performed in parallel, potentially iteratively, with the first bullet point (RGU identification with PM). The MA were identified, delineated, and characterised based on InSAR data (see Section 3.1). For each area, the operators used a similar collection of radar image pairs (interferograms) from different spaceborne radar sensors, with different viewing geometries and variable time intervals between the image acquisitions. In some areas, multi-temporal InSAR mean velocity maps based on Distributed Scatterer (DS) and Persistent Scatterer (PS) algorithms were also available (Table 2). Each recognised MA was delineated in a dedicated polygon vector layer. The attributes documenting the velocity class, the observation time window and validity time frame, and the MA reliability could be filled using a semi-automatic dialog box. The attribute table of the MA layer is shown in Appendix B. The boundaries of the MA polygons follow the InSAR signal, not the landform features. If the movement is heterogenous and/or if InSAR is affected by limitations, the MA may only be partly overlapping with the rock glacier. The MA step was performed before the team decisions on the RGU final locations, which means that some delineated MA may correspond to surface movement associated to uncertain rock glaciers or other periglacial processes. Such polygons were kept in the final layer but were not further used for morpho-kinematic characterisation when they did not correspond to a certain RGU. If no movement was detected on InSAR, no polygon was drawn. Several rock glaciers have therefore no corresponding MA. The complete procedure is explained in the RGIK practical InSAR guidelines (RGIK, 2023b).*

**The case of a rock glacier without any MA is more usual, cause the presence of a rock glacier is not necessarily associated with a detected movement. The absence of detected MA can be either due to 1) no movement (or too low to be detected), or 2) low data quality and/or coverage which did not allow for detecting it. If we can ensure that the absence of movement is caused by 1) a KA category < cm/yr can be assigned and the activity attribute is set to relict based on this information. If we do not know (low or uncertain data quality/coverage), the activity assessment relies on a geomorphological analysis. Such decision is not straightforward and has been identified as a challenge that we need to clarify in the guidelines. See third bullet point in 5.3.2:**

*The assignment of the activity attribute based on geomorphological and/or kinematic criteria requires clarification in the guidelines. The InSAR analysis led to the generation of a MA layer with polygons highlighting where movement has been detected. For characterising the kinematics and the activity, the operators used the MA layer as input. However, some rock glaciers are not covered by any MA. In such cases, it was recommended to avoid overinterpreting the absence of detected movement, because a rock glacier without any MA may mean two different things: 1) there is no movement or too low to be detected, 2) the data quality and/or coverage did not allow for detecting it. In such cases, some operators only focused on geomorphological criteria to assign the activity. A kinematic attribute with low velocity and low reliability index may have been documented, but was not used to set the activity. For other operators, the lack of movement evidence has been used in synergy with geomorphological evidence, as an additional indicator confirming the geomorphological interpretation.*